# Gamma Distribution PCA-Enhanced Feature Learning for Angle-Robust SAR Target Recognition

**Chong Zhang**[1]   **Peng Zhang**[* 1]   **Mengke Li**[* 2]

## Abstract

Scattering characteristics of synthetic aperture radar (SAR) targets are typically related to observed azimuth and depression angles. However, in practice, it is difficult to obtain adequate training samples at all observation angles, which probably leads to poor robustness of deep networks. In this paper, we first propose a Gamma-Distribution Principal Component Analysis ($\Gamma$PCA) model that fully accounts for the statistical characteristics of SAR data. The $\Gamma$PCA derives consistent convolution kernels to effectively capture the angle-invariant features of the same target at various attitude angles, thus alleviating deep models' sensitivity to angle changes in SAR target recognition task. We validate $\Gamma$PCA model based on two commonly used backbones, ResNet and ViT, and conduct multiple robustness experiments on the MSTAR benchmark dataset. The experimental results demonstrate that $\Gamma$PCA effectively enables the model to withstand substantial distributional discrepancy caused by angle changes. Additionally, $\Gamma$PCA convolution kernel is designed to require no parameter updates, introducing no extra computational burden to the network. The source code is available at https://github.com/ChGrey/GammaPCA.

## 1. Introduction

Synthetic aperture radar (SAR), with its all-weather, all-time observation capabilities, plays an important role in various remote sensing observation missions. It has found widespread application in both civil (Yang et al., 2023;

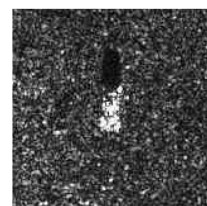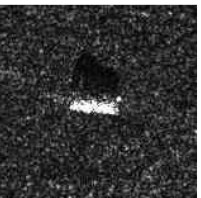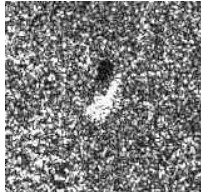

Figure 1: SAR images of the same target captured at different angles. (Left) Azimuth 0°-90°, Depression 17°; (Middle) Azimuth 270°-360°, Depression 17°; (Right) Azimuth 0°-90°, Depression 30°.

Zhang et al., 2023a; Yang et al., 2022; Zhang et al., 2021) and military (Song et al., 2022; Lv et al., 2023; Kechagias-Stamatis & Aouf, 2021) fields. Automatic target recognition (ATR) is a fundamental observation mission of SAR. In the SAR ATR field, numerous studies have attempted to develop general and robust feature extractors by adapting deep networks initially designed for optical images (Pei et al., 2023; Zeng et al., 2022; Guo et al., 2020; Pei et al., 2023; Li et al., 2020; Wang et al., 2021). While these approaches have demonstrated satisfactory performance, they often struggle to maintain robustness when recognizing the same target at different azimuth or depression angles. This is primarily because the electromagnetic scattering characteristics of the same SAR target will vary depending on the angle of incidence of the electromagnetic wave.

In recent years, numerous studies focusing on SAR ATR have been proposed from various perspectives, including electromagnetic scattering feature extraction (Li et al., 2022; Feng et al., 2023; Zhang et al., 2020; Li et al., 2021; Chen et al., 2024; Huang et al., 2024b), phase information (Wang et al., 2022; Liu & Lang, 2021; Zeng et al., 2022), shadow information (Guo et al., 2024) attitude information (Guo et al., 2024; Wang et al., 2024; Oh et al., 2020) as well as methods based on transfer learning (Zhou et al., 2024b; Shi et al., 2024; Zhang et al., 2023c), to name a few. These methods have empirically demonstrated superior performance, thereby emphasizing the importance of accounting for the unique characteristics of SAR data in the SAR ATR task. However, most of physical-based methods rely on extra physical prior knowledge, which limits their generalizability to a wider range of task scenarios. While deep

*Corresponding author. [1]National Key Laboratory of Radar Signal Processing, School of Electronic Engineering, Xidian University, Xi'an, China. [2]College of Computer Science and Software Engineering, Shenzhen University, Shenzhen, China. Correspondence to: Peng Zhang <pzhang@xidian.edu.cn>, Mengke Li <csmengkeli@gmail.com>.

*Proceedings of the 42nd International Conference on Machine Learning*, Vancouver, Canada. PMLR 267, 2025. Copyright 2025 by the author(s).

learning-based approaches mainly consider the problem of data scarcity, and the data shift problem caused by imaging angle variation is usually ignored.

In reality, the distribution of strong scattering points and shadow areas in SAR imagery is highly sensitive to the attitude of a target, that is, the azimuth and depression angle of the target (Dong & Liu, 2022). The same target observed from different angles often exhibits distinct visual appearances, scattering centers, geometric contours, etc., as shown in Figure 1. Therefore, the variation in the scattering characteristics of the same target with azimuth and depression angle is a critical factor affecting the performance of existing methods, yet it has not been explicitly or comprehensively addressed. Although prior studies have attempted to solve this issue based on multiview fusion and azimuth estimation (Liao et al., 2024; Ge et al., 2022; He et al., 2021), these methods are still unable to maintain the robust representation of different inputs from the same class when training samples from multiple angles are lacking. As a result, the sensitivity of target scattering properties to the viewing angle remains challenging in SAR ATR.

To address the aforementioned issue, we propose a feature extractor based on generalized Gamma-distribution principal component analysis (denoted as ΓPCA) to obtain angle-insensitive features. The Gamma distribution is well-suited to model the statistics of multiple independent scattering points, making it particularly effective for SAR targets. This characteristic allows for the extraction of robust low-rank information from targets observed at multiple viewing angles. We extend the Pearson mean square error optimization objective of principal component analysis (PCA) based on the Gamma distribution, resulting in ΓPCA. We further derive the consistency projection matrix, enabling the extraction of low-rank information across different target angles. This approach unifies the intrinsic characteristics of a target across various attitudes, significantly enhancing the ability of models to discriminate and generalize to unseen attitude angles. The proposed ΓPCA is a simple, general, and efficient feature extractor that can be seamlessly integrated with various backbone architectures. We demonstrate the effectiveness of our method by incorporating ΓPCA into both CNN-based and Transformer-based backbones. The experimental results demonstrate that the proposed ΓPCA effectively improves the performance of existing models. The main contributions of this paper are as follows:

- We fully consider the non-Gaussian statistics of SAR targets and derive a novel ΓPCA that can extract low-rank features of SAR targets.

- With SAR target samples observed at finite angles, the proposed ΓPCA derives a projection matrix to construct convolutional kernels, effectively capturing angle-insensitive information.

- The proposed ΓPCA can be seamlessly integrated into various deep models without introducing additional parameters, effectively enhancing the robustness of target recognition across different azimuth and depression angles for SAR data.

## 2. Related Work

### 2.1. Attributed Scattering Center-Based Approaches

The scarcity and attitude sensitivity of SAR data pose significant challenges for achieving robust target recognition (TRR) under limited data conditions. Physical features, known for their robustness and explainability (Datcu et al., 2023), have been widely utilized in addressing these challenges. Among various physical cues, attributed scattering centers (ASC) have emerged as a prominent approach. Early works (Potter & Moses, 1997) demonstrated the effectiveness of ASC in describing local scattering characteristics, leading to the development of robust feature extractors (Ding et al., 2017; 2018) for SAR TRR tasks.

Recent advancements have seen the integration of ASC with deep neural networks (DNNs), evolving from simple feature fusion (Zhang et al., 2020) to more sophisticated ASC-driven models (Zhao et al., 2024; Feng et al., 2022; 2021). These models have shown resilience against significant variations in depression angles (Huang et al., 2024b). Additionally, other physical information such as azimuth, phase, and shadow characteristics have been leveraged to enhance SAR TRR. For example, Zhang et al. (2024) utilize azimuth information to filter and learn salient representations, thereby making the model more robust to azimuth variations. Choi et al. (2022) and Guo et al. (2024) focus on shadow characteristics to guide deep models in learning target structures. Complex-valued networks, integrating amplitude and phase information, have also shown promise in addressing SAR TRR challenges, as demonstrated by Deng et al. (2022); Zeng et al. (2022); Zhou et al. (2023).

Most physics-driven models are inherently task-specific and rely heavily on complex expert knowledge, which limits their generalizability to more refined downstream tasks. Additionally, experiments involving these methods typically address only one specific scenario, such as missing depression or azimuth information, leaving their robustness under more complex and challenging conditions untested.

### 2.2. Deep Model-Driven Approaches

In contrast to physics-driven methods, deep model-driven approaches leverage established deep learning frameworks to address the challenges in SAR TRR. On one hand, a series of studies have developed task-specific deep networks for SAR TRR by utilizing well-established architectures such as ResNet (Shang et al., 2023) and Transformer (Sun et al.,

2024). For instance, Lv et al. (2024) propose a Transformer-based multi-view joint model that addresses challenges related to small target deformation and depression variation by denoising SAR images using low-frequency prior information and integrating multi-view features. On the other hand, various deep learning methods have been applied to SAR TRR tasks. Notably, recent studies have framed the limitations of SAR data as a few-shot learning (FSL) problem and have sought to address it by incorporating diverse FSL methodologies (Zhou et al., 2024b; Zhang et al., 2023c). For example, Zhang et al. (2023b) establish a unified benchmark for few-shot SAR image classification to evaluate the performance of various classical SAR ATR methods. Similarly, Shi et al. (2024) propose an unsupervised domain adaptation approach that employs contrastive learning to model and align the distributions of simulated and real data, thereby mitigating the issue of SAR data scarcity. Deep model-driven approaches exhibit superior generalizability compared to physics-driven approaches. Furthermore, some hybrid methods have emerged that integrate physical information with deep learning frameworks, achieving a balance between the two paradigms (Zhou et al., 2024b; Sun et al., 2024). However, few studies have explored scenarios involving simultaneous variations in both azimuth and depression angles, limiting their applicability in real world.

## 3. Proposed Method

The scattering characteristics of SAR targets are inherently related to the imaging angle. However, acquiring training samples that cover the full range of possible angles is often impractical. As a result, enhancing the robustness of recognition against imaging angle variations remains a significant challenge. Fortunately, we observe that, for a given target, the statistical properties of SAR images exhibit notable consistency despite variations in imaging angles. Figure 2 illustrates this situation and Li et al. (2011) have empirically demonstrated that the distribution of SAR targets follows a gamma distribution. This statistical characteristic forms the basis for our approach to enhance the recognition robustness of targets at unseen angles.

In this section, we propose a Gamma-distributed Principal Component Analysis (PCA) model, named ΓPCA, which can extract low-rank information from SAR targets across multiple viewing angles. ΓPCA extends Pearson's mean square error optimization objective of PCA to the gamma distribution and derives a consistent projection matrix capable of effectively capturing low-rank information under varying target angles, thus achieving robust and accurate target recognition even in scenarios where training samples from different viewing angles are limited. On the other hand, ΓPCA can be seamlessly integrated with various deep learning backbones.

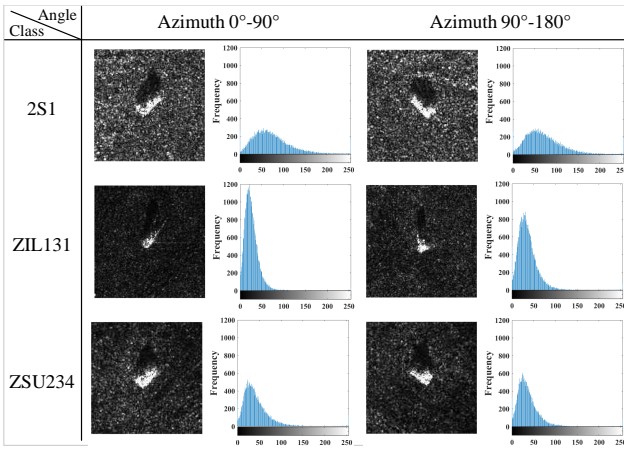

Figure 2: Histogram distribution of three categories of SAR targets at different azimuths.

### 3.1. Gamma-Distributed Principal Component Analysis

To extract low-rank information of targets under varying target angles, we follow the minimum mean square error criterion of generalized PCA (Landgraf & Lee, 2020), where an optimal projection of original $d$ dimensional data $\mathbf{x}_i$ to a $k$ dimensional subspace ($k < d$) is identified with squared error loss. This approach ensures that the projected data retains essential structural information, thereby facilitating feature extraction. Specifically, this projection involves seeking a center $\boldsymbol{\mu} \in \mathbb{R}^d$ and a rank-$k$ matrix $\mathbf{U} \in \mathbb{R}^{d \times k}$, where $\mathbf{U}^T \mathbf{U} = \mathbf{I}_k$ and $\mathbf{I}_k$ is the $k$-dimensional unit matrix, such that the following objective function is minimized:

$$\min \sum_{i=1}^{n} \left\| \mathbf{x}_i - \boldsymbol{\mu} - \mathbf{U}\mathbf{U}^T \left( \mathbf{x}_i - \boldsymbol{\mu} \right) \right\|^2, \text{ s.t. } \mathbf{U}^T \mathbf{U} = \mathbf{I}_k. \tag{1}$$

Traditionally, optimal data representation is derived under the assumption that $x$ follows a Gaussian distribution. However, as discussed above, the SAR target $x$ typically follows a Gamma distribution rather than a Gaussian distribution. In the following, we will discuss how to derive ΓPCA in detail.

**ΓPCA Formulation.** The Gamma distribution is a specific instance of the exponential family of distributions. Generally, the probability density function of the exponential family for a random variable $x$ is expressed as:

$$f(x \mid \theta, \varphi) = \exp \left( \frac{x\theta - b(\theta)}{a(\varphi)} + c(x, \varphi) \right), \tag{2}$$

where $\theta$ is the canonical natural parameter, $a(\varphi)$ is the scale parameter, and $a(\varphi)$, $b(\theta)$, $c(x, \varphi)$ are all determined by specific distribution.

We express the probability density function of Gamma dis-

tribution into the form of exponential family distribution:

$$f(x\,|\,u,v) = \frac{u^v}{\Gamma(v)} x^{v-1} e^{-ux}$$
$$= \exp\{v \cdot \log u - \log \Gamma(v) + (v-1)\log x - ux\}$$
$$= \exp\left\{ \frac{-\frac{ux}{v} + \log u}{v^{-1}} - \log \Gamma(v) + (v-1)\log x \right\}.$$
$$(3)$$

In the above equation, $\Gamma(v) = \int_0^\infty x^{v-1} e^{-x} dx$. If we assume that $-\frac{u}{v} = \theta$, $v^{-1} = \varphi^2$, Equation (3) can be reformulated as

$$f(x\,|\,\theta,\varphi) = \exp\left\{ \begin{array}{c} \frac{x\theta + \log(-\theta)}{\varphi^2} + \frac{1}{\varphi^2}\log\frac{1}{\varphi^2} \\ -\log\Gamma\left(\frac{1}{\varphi^2}\right) + \left(\frac{1}{\varphi^2} - 1\right)\log x \end{array} \right\}.$$
$$(4)$$

By comparing Equation (2) with Equation (4), the undetermined functions of Gamma distribution are as follows:

$$a(\varphi) = \varphi^2,$$
$$b(\theta) = -\log(-\theta),$$
$$c(x,\varphi) = \frac{1}{\varphi^2}\log\frac{1}{\varphi^2} - \log\Gamma\left(\frac{1}{\varphi^2}\right) + \left(\frac{1}{\varphi^2} - 1\right)\log x.$$
$$(5)$$

According to the property of exponential family distribution (Landgraf & Lee, 2020), the mean of Gamma distribution can be directly derived from:

$$E(X) = b'(\theta) = -\theta^{-1}. \qquad (6)$$

In addition, the saturated model is considered (Landgraf & Lee, 2020), and we have a canonical link function $g(\cdot)$ which satisfies $g(b'(\theta)) = \theta$ and the mean $b'(\tilde{\theta}) = x$. This also implies that $g(b'(\tilde{\theta})) = g(x) = \tilde{\theta}$. Therefore, we can approximate the natural parameter $\theta$ in $\Gamma$PCA by:

$$b'(\tilde{\theta}) = -\tilde{\theta}^{-1} = x, \text{ i.e. } \tilde{\theta} = -x^{-1}. \qquad (7)$$

Optimal data representation in $\Gamma$PCA is achieved when the estimated mean vector $\hat{\boldsymbol{\mu}}$ equals the sample mean $\bar{\mathbf{x}}$ and the estimated subspace $\hat{\mathbf{U}}$ consists of the first $k$ eigenvectors of the sample covariance matrix. Therefore, the approximation of $\mathbf{x}_i$ by Equation (1) can be viewed equivalently as the approximation of $\tilde{\theta}_{ij}$ by

$$-\hat{\theta}_{ij}^{-1} = \mu_j + [\mathbf{U}\mathbf{U}^T(-\tilde{\boldsymbol{\theta}}_i^{-1} - \boldsymbol{\mu})]_j, \qquad (8)$$

where $\tilde{\boldsymbol{\theta}}_i$ is the natural parameter for the $i$-th case of saturated model. For similarity, we denote that $\tilde{\boldsymbol{\eta}}_i(\theta) = [\tilde{\theta}_{i1}^{-1}, \tilde{\theta}_{i2}^{-1}, \ldots, \tilde{\theta}_{id}^{-1}]^T$, and the estimation of natural parameter $\hat{\theta}_{ij}$ in $k$-dimensional subspace can be defined as

$$\hat{\theta}_{ij} = \left( [\mathbf{U}\mathbf{U}^T(\tilde{\boldsymbol{\eta}}_i(\theta) + \boldsymbol{\mu})]_j - \mu_j \right)^{-1}. \qquad (9)$$

**$\Gamma$PCA Objective.** By Equation (9), we can derive the approximation form $\hat{\theta}_{ij}$ of the natural parameter $\theta_{ij}$ in $\Gamma$PCA.

Then, how to derive the rank-$k$ matrix $\mathbf{U} \in \mathbb{R}^{d \times k}$ and the center $\boldsymbol{\mu} \in \mathbb{R}^d$? To derive them, we minimize the deviance:

$$D(\mathbf{X}; \hat{\boldsymbol{\Theta}}) = -2\sum_{i=1}^n \sum_{j=1}^d (\log f(x\,|\,\hat{\theta}, \varphi) - \log f(x\,|\,\tilde{\theta}, \varphi)).$$
$$(10)$$

where the data matrix $\mathbf{X} = [x_{ij}]$ and the natural parameter matrix $\hat{\boldsymbol{\Theta}} = [\theta_{ij}]$.

Substituting Equation (4) into Equation (10), and further considering $\tilde{\theta} = -x^{-1}$, we derive the following formula:

$$\sum_{i=1}^n \sum_{j=1}^d D_j(x_{ij}; \hat{\theta}_{ij}) \propto \sum_{i=1}^n \sum_{j=1}^d \{-x_{ij}\hat{\theta}_{ij} - \log(-\hat{\theta}_{ij})\}.$$
$$(11)$$

This formula can also be seen as minimizing the deviance of natural parameters in the $k$-dimensional subspace. Substituting Equation (9) into Equation (11), the optimization objective function can be written as

$$\max_{\substack{\boldsymbol{\mu} \in \mathbb{R}^d \\ \mathbf{U}^T\mathbf{U} = \mathbf{I}_k}} \sum_{i,j} \left\{ \begin{array}{l} x_{ij}\left([\mathbf{U}\mathbf{U}^T(\tilde{\boldsymbol{\eta}}_i(\theta) + \boldsymbol{\mu})]_j - \mu_j\right)^{-1} \\ -\log\left(\mu_j - [\mathbf{U}\mathbf{U}^T(\tilde{\boldsymbol{\eta}}_i(\theta) + \boldsymbol{\mu})]_j\right) \end{array} \right\}.$$
$$(12)$$

Hence, by minimizing the the objective function in Equation (12), $\Gamma$PCA finds a center $\boldsymbol{\mu} \in \mathbb{R}^d$ and a rank-$k$ matrix $\mathbf{U} \in \mathbb{R}^{d \times k}$ such that $\mathbf{U}^T\mathbf{U} = \mathbf{I}_k$.

The above objective function is generally non-convex, making it challenging to find the global optimal solution. Therefore, we iteratively optimize the objective function using the Majorization-Minimization (MM) algorithm (Mairal, 2013; Sun et al., 2016; Landgraf & Lee, 2020). Then, $\mathbf{U}$ and $\boldsymbol{\mu}$ are estimated iteratively by:

$$\boldsymbol{\mu}^{(t+1)} = (\mathbf{1}_n^T\mathbf{Q}^{(t)}\mathbf{1}_n)^{-1}\left(\mathbf{H}\mathbf{U}^{(t)}(\mathbf{U}^{(t)})^T - \mathbf{P}^{(t)}\right)^T \mathbf{Q}^{(t)}\mathbf{1}_n.$$
$$(13)$$

$$\mathbf{F}^{(t)} = \left(\mathbf{H} + \mathbf{1}_n(\boldsymbol{\mu}^{(t)})^T\right)^T \mathbf{Q}^{(t)}\left(\mathbf{P}^{(t)} + \mathbf{1}_n(\boldsymbol{\mu}^{(t)})^T\right)$$
$$+ \left(\mathbf{P}^{(t)} + \mathbf{1}_n(\boldsymbol{\mu}^{(t)})^T\right)^T \mathbf{Q}^{(t)}\left(\mathbf{H} + \mathbf{1}_n(\boldsymbol{\mu}^{(t)})^T\right) \quad (14)$$
$$- \left(\mathbf{H} + \mathbf{1}_n(\boldsymbol{\mu}^{(t)})^T\right)^T \mathbf{Q}^{(t)}\left(\mathbf{H} + \mathbf{1}_n(\boldsymbol{\mu}^{(t)})^T\right),$$

where $\mathbf{U}^{(t+1)}$ is the first $k$ eigenvectors of $\mathbf{F}^{(t)}$.

The detailed derivation of Equation (13) and Equation (14) are provided in Appendix B. In the MM algorithm, $\boldsymbol{\mu}^{(1)}$ is initialized as the column-wise mean vector of $\tilde{\boldsymbol{\Theta}}$, and $\mathbf{U}^{(1)}$ is initialized using the first $k$ right singular vectors of $\tilde{\boldsymbol{\Theta}}$. The procedure of $\Gamma$PCA is summarized in Algorithm 1.

### 3.2. $\Gamma$PCA Attitude-Robust Convolution Kernel

In the preprocessing of original SAR images, following the approach of PCANet (Chan et al., 2015), we divide each

---

**Algorithm 1** ΓPCA Algorithm.

---

**Require:** Input data $\mathbf{X} = (\mathbf{x}_1, \mathbf{x}_2, \cdots, \mathbf{x}_N)$, number of extracted principal components $L$, size of projection matrix $k$, number of iterations for MM algorithm $T$.

**Ensure:** Output orthonormal optimal projection matrix $\mathbf{V}$

1: **Initialize:** $\boldsymbol{\mu}^{(0)}$ to the column means of $\tilde{\boldsymbol{\Theta}}$ and $\mathbf{U}^{(0)}$ to the first $k$ right singular vectors of $\tilde{\boldsymbol{\Theta}}$, where $\boldsymbol{\Theta}$ is calculated by Equation (7).

2: **for** $t = 0$ **to** $T$ **do**

3:    Update the mean vector $\boldsymbol{\mu}^{(t+1)}$ by $\boldsymbol{\mu}^{(t+1)} = (\mathbf{1}_n^T \cdot \mathbf{Q}^{(t)} \mathbf{1}_n)^{-1} \Big( \mathbf{H}\mathbf{U}^{(t)} (\mathbf{U}^{(t)})^T - \mathbf{P}^{(t)} \Big)^T \mathbf{Q}^{(t)} \mathbf{1}_n$.

4:    Calculate and record the deviance $M(\boldsymbol{\Theta} \,|\, \boldsymbol{\Theta}^{(t)})$ by

$$M^{(t)} = \left\| \begin{array}{c} \left(\mathbf{Q}^{(t)}\right)^{\frac{1}{2}} \left(\mathbf{H} + \mathbf{1}_n (\boldsymbol{\mu}^{(t)})^T\right) \mathbf{U}^{(t)} \left(\mathbf{U}^{(t)}\right)^T \\ - \left(\mathbf{Q}^{(t)}\right)^{\frac{1}{2}} \left(\mathbf{P}^{(t)} + \mathbf{1}_n (\boldsymbol{\mu}^{(t)})^T\right) \end{array} \right\|_F^2.$$

5:    **if** $M^{(t)} \leq \min[M^{(k)}]_{k=1}^t$ **then**

6:       $\mathbf{V} \leftarrow \mathbf{U}^{(t)}$.

7:    **end if**

8:    Update the mean vector $\mathbf{U}^{(t+1)}$ ( by Equation (29) in Appendix B.2), where $\mathbf{U}^{(t+1)}$ is the first $k$ eigenvectors of $\mathbf{F}^{(t)}$.

9: **end for**

---

SAR image into multiple patches, which are then vectorized for further processing. Suppose ΓPCA model receives a data set with $N$ images, where the optimal projection matrix has dimensions $k \times k$, and the goal is to extract $L$ principal components from each picture. Initially, input images should be reshaped to a uniform size, and we denoted the reshaped images as $\mathbf{I} = [\mathbf{I}_i]_{i=1}^N \in \mathbb{R}^{m \times m}$. Then, all the images are padded with blank pixels and divided into $\tilde{m}^2$ patches of size $k \times k$. For the $i$-th image, we vectorize its all patches and concatenate them to a new matrix $\mathbf{P}_i \in \mathbb{R}^{k^2 \times \tilde{m}^2}$, where $\tilde{m} = \lfloor (m - k + 2p)/s \rfloor + 1$, $p$ is the number of blank pixels padded around a image, and $s$ is the stride size of dividing patches.

Subsequently, we calculate the mean matrix over $[\mathbf{P}_i]_{i=1}^N$ and derive its second-order statistics, which serve as input to principal component analysis:

$$\mathbf{X} = \frac{1}{N\tilde{m}^2} \mathbf{P}\mathbf{P}^T, \qquad (15)$$

where $\mathbf{P} = \sum_{i=1}^N \mathbf{P}_i$. Additionally, we divide the result by $\tilde{m}^2$ to ensure that each matrix element falls within a range of 0 to 1, thereby facilitating the calculation of ΓPCA.

By the processing shown in Algorithm 1, we can finally get an orthonormal optimal projection matrix $\mathbf{V} \in \mathbb{R}^{k^2 \times L}$. Then $\mathbf{V}$ can be reshaped into $L$ convolution kernels of size $k \times k$.

$$\mathbf{W}_l = \mathrm{mat}_{k \times k} (\mathbf{V}_{:,l}), l = 1, 2, \cdots, L, \qquad (16)$$

| Class Image | 2S1 (Depression 15°) | 2S1 (Depression 30°) | ZSU234 (Depression 15°) | ZSU234 (Depression 30°) |
|---|---|---|---|---|
| Original SAR image | | | | |
| 1st Principal Component $\boldsymbol{O}_i^1$ | | | | |
| 2nd Principal Component $\boldsymbol{O}_i^2$ | | | | |

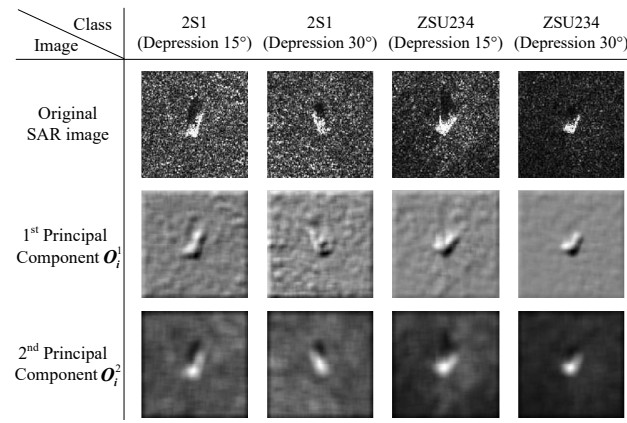

Figure 3: ΓPCA feature visualization.

where $\mathrm{mat}_{k \times k} (\mathbf{V}_{:,l})$ denotes that mapping the $l$-th column of matrix $\mathbf{V}$ to a new matrix $\mathbf{W}_l \in \mathbb{R}^{k \times k}$.

The optimal projection matrix can be regarded as a conventional convolution kernel to convolve with the original SAR images:

$$\mathbf{O}_i^l = \mathbf{W}_l * \mathbf{I}_i \in \mathbb{R}^{\tilde{m} \times \tilde{m}}, i = 1, 2, \cdots, N, l = 1, 2, \quad (17)$$

where $\mathbf{O}_i^l$ denotes the $l$-th principal component of $i$-th image, which has a size of $\tilde{m} \times \tilde{m}$. Figure 3 illustrates the feature extraction performance of ΓPCA convolution kernels trained with limited angles while applied to samples under unseen angles.

ΓPCA functions as a filter within deep networks, facilitating more effective preservation of angle-invariant features for the same SAR targets across different attitude angles, thereby enhancing the robustness of the networks. The implementation process of ΓPCA is illustrated in Figure 4.

In this paper, each SAR image is decomposed into two principal components, denoted as $\mathbf{O}_i^1$ and $\mathbf{O}_i^2$. These principal component feature maps retain the same spatial dimensions as the original SAR image, enabling them to be concatenated with the single-channel image along the channel dimension. The resulting three-channel image, comprising the two feature maps and the original image, serves as input for subsequent deep network processing.

Notably, in ΓPCA, we employ Min-Max Scaling for image normalization instead of conventional normalization techniques such as LayerNorm, BatchNorm, or InstanceNorm. It is motivated by two key considerations: 1) Preservation of principal component independence: In SAR images, each feature map corresponds to a single principal component of the target, and these components exhibit significant diversity. To maintain the independence between principal components of different styles, it is crucial to avoid normalization methods that might introduce unintended dependencies. 2) Distributional compatibility: LayerNorm, BatchNorm,

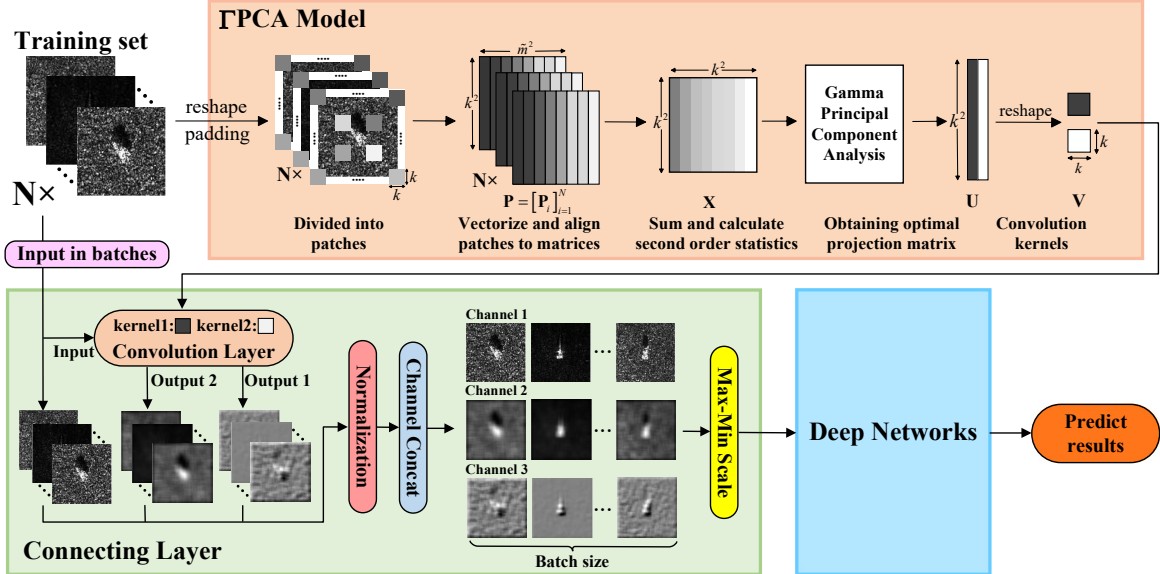

Figure 4: Implementation of Gamma-Distribution Principal Analysis (ΓPCA).

and InstanceNorm are designed under the assumption that the data follows a Gaussian distribution. However, SAR data inherently follow a Gamma distribution, and applying Gaussian-based normalization methods could distort the intrinsic distributional characteristics of the data. Min-Max scaling normalizes data by scaling each channel of an image to the range $[-1, 1]$ based on its maximum and minimum values. This approach is distribution-correlated, preserving the inherent structure of SAR data without imposing Gaussian assumptions.

## 4. Empirical Study

### 4.1. Description of MSTAR Dataset

The data set used in this paper is the static ground military target data set, MSTAR (Keydel et al., 1996). MSTAR is collected by a high resolution beam stacking SAR. It consists of various vehicle images at different azimuth and depression angles. The detailed composition of MSTAR dataset is illustrated in Appendix C.1.

### 4.2. Comparison Methods and Experimental Settings

**Comparison Methods.** To demonstrate the effectiveness and generalizability of the proposed method, we integrate ΓPCA into two representative backbones, ResNet101 (He et al., 2016) and ViT-B/16 (Dosovitskiy, 2020), referred to as ΓPCA-ResNet and ΓPCA-ViT, respectively. In addition to the baseline modelsFurthermore, w namely ResNet101 and ViT-B/16, We compare our method with three recently proposed backbones: Swin Transformer (Liu et al., 2021), EfficientNet (Tan & Le, 2019), and VisionMamba (Zhu et al., 2024). Furthermore, we evaluate our method against sev-

eral SOTA approaches, including VisionLSTM-T (Alkin et al., 2024), MSNet-PIHA (Huang et al., 2024b), and WTConvNeXt-T (Finder et al., 2025).

**Settings.** For the dataset, in previous studies, experimental settings typically considered variations in either azimuth or depression angle individually. However, in real-world scenarios, limited azimuth coverage and changes in depression angle often occur simultaneously. To better evaluate model robustness under such realistic conditions, we design a more challenging experimental setting that incorporates both azimuth insufficiency and depression variation. Our experimental design is as follows.

*Azimuth Robustness Test.* To simulate the scenario that a wide range of azimuth angles are missing, all models are trained and validated by data from only one azimuth quadrant at depression 17° (e.g., Azimuth 0°-90°, Depression 17°), which means these data only consist of 25 % azimuth information at that depression angle. For testing, a well-trained model will be tested in the full-azimuth (0°-360°) testing set at depression 15°.

*Azimuth & Depression Robustness Test.* The processes of model training and validation are the same as Azimuth Robustness Test. However, all trained models will be tested in a testing set at depression 30°, full-azimuth.

To reduce the randomness, we conducted the above experiments on two separate training sets, which come from opposite azimuth quadrants: 0°-90° and 180°-270°. It is generally believed that two opposite azimuth quadrants can provide complementary azimuthal information, therefore, we would like to repeatedly evaluate a model's stability by using two complementary training sets. A detailed description of the experimental strategy is provided in Appendix A.

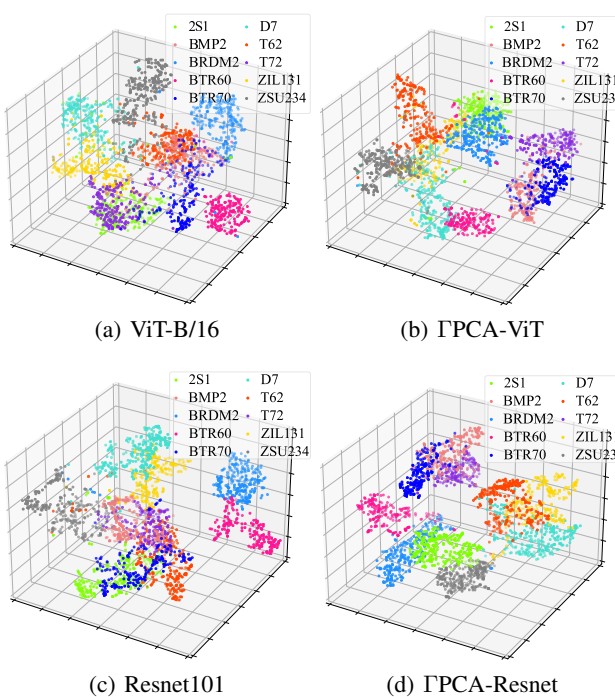

(a) ViT-B/16        (b) ΓPCA-ViT

(c) Resnet101       (d) ΓPCA-Resnet

Figure 5: t-SNE visualizations of feature distributions for different backbones in A-R Test. Training set: Depression 17°, Azimuth 0°-90°.

For the backbone, due to the small size of our dataset, all models are pre-trained on ImageNet and then fine-tuned on our dataset. All networks only use Resize and CenterCrop to preprocess the input data. For the hyperparameters, the ΓPCA part uses $L = 2$ kernels with a kernel size of $k = 17$.

### 4.3. Results on MSTAR dataset

We report the test accuracy in Table 1 and Table 2. Through an overall comparison, ΓPCA-ViT achieves an overall accuracy of 76.02% and 75.22% on the two separate training sets, respectively, with both being the best performance among all models. The results demonstrate the superiority and stability of ΓPCA-ViT.

In terms of individual tests, ΓPCA-ViT consistently maintains leading performances on A-R Test even with different training sets. On AD-R Test, it is noticeable that the performance of most models has dropped significantly, which is consistent with the previous discussion of the attitude sensitivity effect. Depression angle is a crucial factor that affects the target's reflection intensity and shadow shape. Thus, its variations generally cause a structure distribution shift of the dataset. Although the performances of ΓPCA-ViT on AD-R Test are not leading, we can find that models with high accuracy on AD-R Test always perform poorly on A-R Test, for example, Vision-LSTM and VisionMamba, which are not considered to be robust to various attitude variations. Our comparative analysis indicates that ViT performs robustly

Table 1: Recognition results of different networks. Training set: Depression 17°, Azimuth 0°-90°; Testing set: Depression 15°, Full-Azimuth & Depression 30°, Full-Azimuth. OA is short for Overall Accuracy.

| Methods | A-R Test | AD-R Test | OA (%) |
|---|---|---|---|
| Resnet101 | 72.57±0.25 | 43.95±0.01 | 63.62±0.14 |
| EfficientNet-B3 | 67.22±0.02 | 45.15±0.07 | 60.27±0.01 |
| ViT-B/16 | 78.12±0.04 | 60.11±0.24 | 72.45±0.04 |
| Swin-B | 61.58±0.03 | 78.01±0.04 | 66.75±0.02 |
| VisionLSTM-T | 55.25±0.21 | 76.54±0.24 | 61.95±0.22 |
| VisionMamba-T | 62.37±0.06 | 66.67±0.03 | 63.68±0.04 |
| MSNet-PIHA | 68.08±0.03 | 55.75±0.06 | 64.20±0.03 |
| WTConvNeXt-T | 72.55±0.01 | 57.69±0.05 | 67.87±0.01 |
| ΓPCA-Resnet (Ours) | 73.79±0.02 | 52.96±0.05 | **67.23±0.01** |
| ΓPCA-ViT (Ours) | 78.48±0.01 | 70.67±0.16 | **76.02±0.01** |

Table 2: Recognition results of different networks. Training set: Depression 17°, Azimuth 180°-270°; Testing set: Depression 15°, Full-Azimuth & Depression 30°, Full-Azimuth. OA is short for Overall Accuracy.

| Methods | A-R Test | AD-R Test | OA (%) |
|---|---|---|---|
| Resnet101 | 75.30±0.01 | 48.12±0.68 | 66.74±0.11 |
| EfficientNet-B3 | 69.24±0.09 | 45.69±0.07 | 61.83±0.07 |
| ViT-B/16 | 79.92±0.04 | 58.56±0.12 | 73.19±0.01 |
| Swin-B | 70.17±0.08 | 65.53±0.05 | 68.71±0.06 |
| VisionLSTM | 55.49±0.13 | 74.03±1.26 | 61.33±0.16 |
| VisionMamba-T | 66.21±0.02 | 72.98±0.02 | 68.34±0.01 |
| MSNet-PIHA | 65.36±0.02 | 56.55±0.11 | 62.59±0.03 |
| WTConvNeXt-T | 76.46±0.07 | 63.29±0.51 | 72.32±0.16 |
| ΓPCA-Resnet (Ours) | 71.34±0.04 | 66.21±0.02 | **69.72±0.03** |
| ΓPCA-ViT (Ours) | 80.15±0.01 | 64.48±0.05 | **75.22±0.01** |

and maintains stable performance across different scenarios.

Furthermore, vertical comparisons show substantial improvements in model robustness when integrated with ΓPCA. The overall accuracy of ΓPCA-ViT increases by 3.57% and 2.03% over the original ViT-B/16 on the training set 0°-90° and 180°-270°, respectively. Similarly, ΓPCA-ResNet improves on the original ResNet101 by 3.61% and 2.98% on the training set 0°-90° and 180°-270°. In particular, the improvements on the AD-R Test for both ΓPCA-ViT and ΓPCA-ResNet are particularly significant. Under more severe depression variation, the enhanced representations of the target extracted by ΓPCA help the models better adapt their learned patterns to the distribution shift in the test set. These results collectively demonstrate the effectiveness and general applicability of our method across different backbone architectures.

In order to visually see the impact of adding ΓPCA on the feature extraction ability of models, we report the t-SNE visualizations of feature distributions for ViT-B/16, ΓPCA-

Table 3: Recognition results of different networks. Training set: Depression 17°, Azimuth 0°-90°; Testing set: Depression 15°, Full-Azimuth & Depression 30°, Full-Azimuth.

| Methods | A-R Test | AD-R Test | Overall Acc. (%) |
|---|---|---|---|
| Kernel size = 5 | | | |
| ViT-B/16 (Baseline) | 77.98 | 54.58 | 70.61 |
| Conv-ViT | 69.69 | 63.73 | 67.82 |
| WT-ViT | 71.63 | 57.27 | 67.11 |
| PCA-ViT | 78.35 | 52.33 | 70.16 |
| ΓPCA-ViT (Ours) | 76.58 | 63.82 | **72.56** |
| Kernel size = 11 | | | |
| ViT-B/16 (Baseline) | 77.98 | 54.58 | 70.61 |
| Conv-ViT | 65.61 | 68.31 | 66.46 |
| WT-ViT | 69.03 | 68.04 | 68.72 |
| PCA-ViT | 76.08 | 61.67 | 71.55 |
| ΓPCA-ViT (Ours) | 79.13 | 61.40 | **73.55** |
| Kernel size = 17 | | | |
| ViT-B/16 (Baseline) | 77.98 | 54.58 | 70.61 |
| Conv-ViT | 66.39 | 53.50 | 62.33 |
| WT-ViT | 71.51 | 68.22 | 70.47 |
| PCA-ViT | 78.06 | 63.64 | 73.52 |
| ΓPCA-ViT (Ours) | 79.88 | 67.59 | **76.01** |

ViT, Resnet and ΓPCA-ViT in A-R Test, as shown in Figure 5. It is evident that both ΓPCA-ViT and ΓPCA-Resnet have more compact feature distributions than their original counterparts, consistently across nearly all categories. The clustering result verifies that ΓPCA can retain targets' main characteristics and extract effective features to discriminate different categories of targets.

Furthermore, we use different kinds of convolutional layers to replace our ΓPCA layer in ΓPCA-ViT to compare their performances, including a conventional convolutional layer, a standard PCA convolutional layer and a Wavelet convolutional layer (Finder et al., 2025). We temporarily denote them as Conv-ViT, PCA-ViT, and WT-ViT, respectively, for clarity in the following discussions. These convolutional layers all maintain the same formulation of outputs as ΓPCA model. In particular, the standard PCA convolution kernel is built by using the same framework as our ΓPCA model, but its weights are derived through conventional PCA processing. The experimental results are reported in Table 3.

The comparison in Table 3 shows that ΓPCA-ViT consistently achieves prominent performance across different kernel sizes. Besides, adding an extra convolutional layer tends to degrade the performance of ViT in most cases, regardless of whether small or large kernels are used. However, this trend does not hold to the standard PCA and ΓPCA variants. In fact, both PCA-ViT and ΓPCA-ViT exhibit improved performance as the kernel size increases, which aligns with the design principles underlying their convolutional kernels.

Instead of directly performing ΓPCA on the whole SAR image, we split the image into overlapping patches and vec-

Table 4: Recognition results of different networks on SAR-AIRcraft-1.0 dataset.

| Methods | Overall Acc. (%) |
|---|---|
| Resnet101 | 97.33±0.02 |
| ΓPCA+Resnet101 | **98.67±0.01** |
| ViT-B/16 | 97.05±0.03 |
| ΓPCA+ViT-B/16 | **98.12±0.02** |
| Swin-B | 97.41±0.01 |
| ΓPCA+Swin-B | **98.32±0.02** |

torize each patch as a dimension for dimensionality reduction. We expect ΓPCA to consider more target information rather than background information. This processing effectively avoids Γ PCA from capturing the variance of the entire dimension of the image, and leads to a more accurate estimation of the distribution parameters of a target during dimension reduction. In addition, this is also the reason why a larger kernel has a better effect for both ΓPCA and standard PCA. Because a smaller kernel can only make patches cover local parts of a target, which makes the subsequent PCA processing has a large deviation for the estimation of target distribution characteristics.

To further observe the difference between standard PCA and ΓPCA, we have removed the subsequent deep network and only used a PCA convolutional layer and a ΓPCA convolutional layer to extract features of original SAR images respectively, and then obtained their feature visualizations by t-SNE. The results are shown in Figure 6. Obviously, standard PCA is far less effective in characterizing the features of SAR compared to Γ PCA. This is largely because the Gaussian assumption of standard PCA does not fit the distribution characteristics of the SAR data, leading to inadequate separability between classes.

### 4.4. Results on Additional Dataset

MSTAR is a typical dataset for SAR ATR task, and most studies (Huang et al., 2024b; Zhang et al., 2024; Wang et al., 2024; Li et al., 2023) rely solely on MSTAR as their experimental dataset. To further evaluate the generality of our method, we construct a new dataset from the widely used SAR aircraft target detection dataset, SAR-AIRcraft-1.0. We crop every single target from the original SAR image and obtain an available SAR ATR dataset. A detailed description of this dataset is provided in Appendix C.2.

Since SAR-AIRcraft-1.0 lacks diverse and separable imaging angles (Zhirui et al., 2023; Zhou et al., 2024a; Huang et al., 2024a), we employ a standard target recognition strategy and conduct experiments accordingly. Specifically, 80% of the dataset is allocated for training and the remaining 20% for testing. The corresponding results are reported in Table 4. Unlike the results on the MSTAR dataset, several

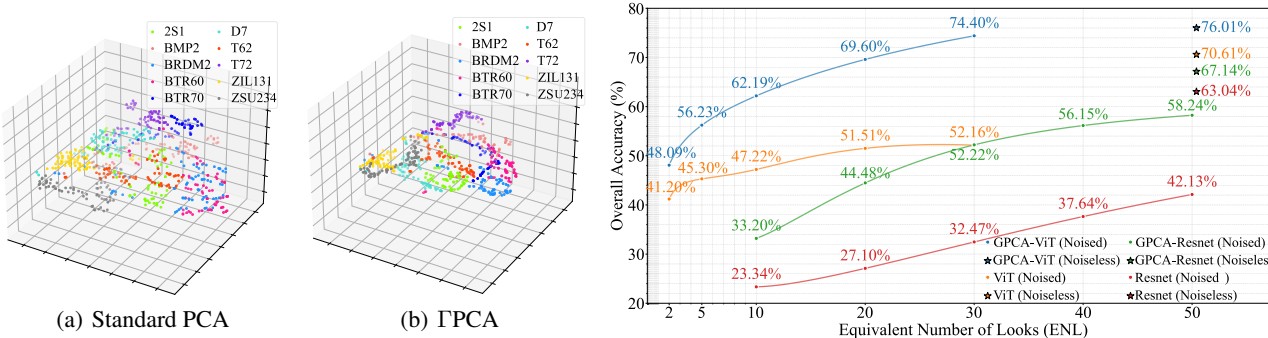

Figure 6: Standard PCA vs. ΓPCA for feature extraction.

Figure 7: Anti-noise performance comparison.

methods achieve satisfying performance on the new dataset. This is because SAR-AIRcraft-1.0 does not include imaging angle annotations, making the task not angle-limited and inherently less complicated. Nevertheless, methods integrated with ΓPCA still outperform their original backbones, further validating the effectiveness and generalizability of our approach across different datasets and conditions.

### 4.5. Anti-Noise Experiment

Compared to optical imagery, SAR images are considerably more susceptible to noise. Among various types of noise, speckle is the most prevalent and detrimental in SAR data. It is a form of inherent multiplicative noise due to the coherent imaging mechanism of SAR, typically manifesting itself as random granular fluctuations across the image. Owing to its strong coupling with the underlying signal, speckle is notoriously difficult to suppress. It can frequently cover weak target information, such as dark shadows or small objects, and severely degrade critical image features like edges and textures, thereby affecting the performance of downstream processing tasks. Consequently, anti-noise performance evaluation is indispensable for ensuring the reliability and effectiveness of SAR ATR methods.

We conduct anti-noise experiments under various speckle conditions. Noise levels are quantified using a standard criterion for multiplicative noise: the Equivalent Number of Looks (ENL), defined as $\mu^2/\sigma^2$, where $\mu$ denotes the mean pixel value and $\sigma$ represents the standard deviation. A higher ENL value indicates lower noise and vice versa. The effects of varying noise levels on different models are illustrated in Figure 7. As shown in the figure, both ViT-B/16 and ResNet101 integrated with ΓPCA exhibit superior anti-noise performance compared to their original counterparts. This improvement is primarily due to the ability of ΓPCA to extract principal components while suppressing subsidiary components, which are dominated by noise. Such selective retention is crucial for enhancing the dual robustness of our method against both angular variations and speckle.

## 5. Conclusion

In the practical SAR target recognition task, data scarcity poses a significant challenge. To address this issue, we have proposed ΓPCA, which effectively improves the robustness of recognition models when the angle information of training data is insufficient. Moreover, ΓPCA is designed as a plug-and-play module, enabling broad applicability across various deep learning architectures. The core idea of ΓPCA is to provide low-rank estimates of natural parameters by projecting saturated model parameters, thereby mining effective low-dimensional features. This approach makes the principal components of the raw data more salient, allowing the network to directly learn essential features of the same target across different imaging angles, significantly mitigating the impact of angle variations. The effectiveness and generalizability of our method have been demonstrated through different angle-insufficient experiments.

## Acknowledgements

This work was supported in part by the Foundation of National Key Laboratory of Radar Signal Processing under Grant (KGJ202304, JKW202403), NSFC (62306181), and Guangdong Basic and Applied Basic Research Foundation (2024A1515010163).

## Impact Statement

This paper aims to advance automatic target recognition for SAR imagery and machine learning. The proposed methods may have implications for military applications, such as target recognition, threat detection, and surveillance. While these technologies could enhance situational awareness and operational efficiency, we acknowledge their dual-use nature and emphasize the importance of responsible deployment. We encourage the adoption of ethical guidelines and policy frameworks to maximize societal benefit while minimizing unintended consequences.

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

## A. Supplementary Description of Experiments

The data are divided into the training set and validation set with a ratio of 0.8. The visualization of the experimental strategy is shown in Figure 8.

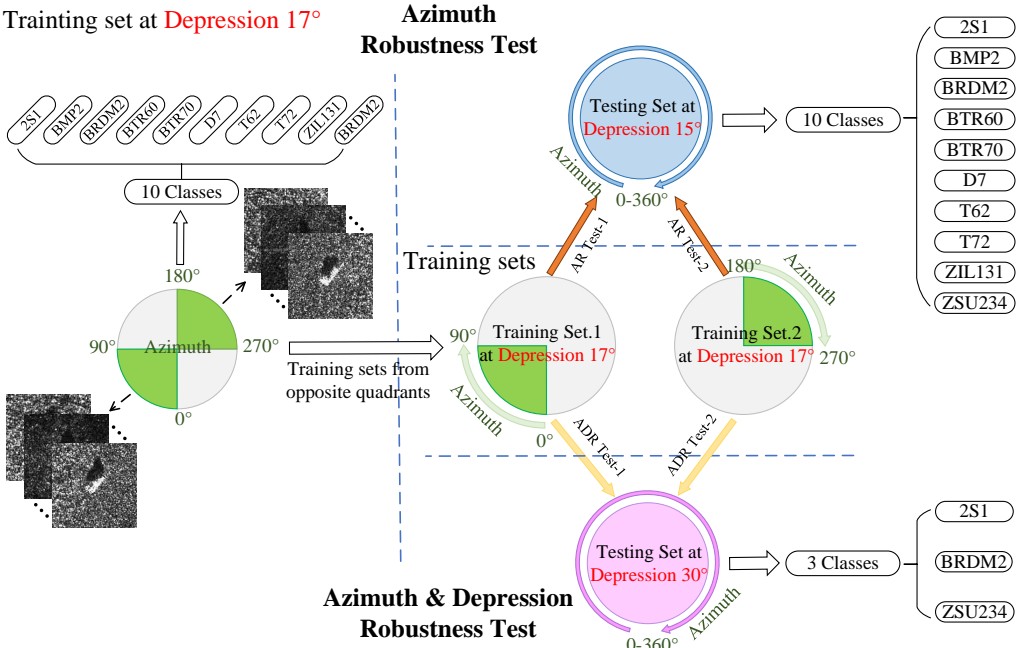

Figure 8: Experimental strategy.

## B. Supplementary Derivation

### B.1. Solvable convex optimization objective

We suppose that $O(\hat{\theta}_{ij} \mid \theta_{ij}^{(t)})$ is the quadratic approximation of deviance $D(x_{ij} \mid \theta_{ij}^{(t)})$ at the point $\theta_{ij}^{(t)}$, and it is defined as

$$
\begin{aligned}
O(\hat{\theta}_{ij} \mid \theta_{ij}^{(t)}) :=& D(x_{ij}; \theta_{ij}^{(t)}) \\
&+ 2w_{ij}\left(b_j'(\theta_{ij}^{(t)}) - x_{ij}\right)\left(\hat{\theta}_{ij} - \theta_{ij}^{(t)}\right) \\
&+ w_{ij} b_j''(\theta_{ij}^{(t)})\left(\hat{\theta}_{ij} - \theta_{ij}^{(t)}\right)^2,
\end{aligned}
\tag{18}
$$

where $t$ refers to the number of current iteration.

As discussed above, we have $b(\theta) = -\log(-\theta)$ for data in the Gamma distribution, and the quadratic approximation $O(\hat{\theta} \mid \theta^{(t)})$ can be replaced as:

$$
\begin{aligned}
O(\hat{\theta}_{ij} \mid \theta_{ij}^{(t)}) :=& D(x_{ij}; \theta_{ij}^{(t)}) \\
&+ 2w_{ij}\left(-1/\theta_{ij}^{(t)} - x_{ij}\right)\left(\hat{\theta}_{ij} - \theta_{ij}^{(t)}\right) \\
&+ w_{ij}/\left(\theta_{ij}^{(t)}\right)^2\left(\hat{\theta}_{ij} - \theta_{ij}^{(t)}\right)^2.
\end{aligned}
\tag{19}
$$

Since $\theta_{ij}^{(t)}$ is not constant, we have the following expression:

$$
v_i^{(t)} := \max_j [w_{ij}/(\theta_{ij}^{(t)})^2].
\tag{20}
$$

The objective function of MM algorithm is expressed by:

$$
\begin{aligned}
M(\hat{\theta}_{ij} \mid \theta_{ij}^{(t)}) :=& D(x_{ij}; \theta_{ij}^{(t)}) \\
&+ 2w_{ij}\left(-1/\theta_{ij}^{(t)} - x_{ij}\right)\left(\hat{\theta}_{ij} - \theta_{ij}^{(t)}\right) \\
&+ v_i^{(t)}\left(\hat{\theta}_{ij} - \theta_{ij}^{(t)}\right)^2.
\end{aligned}
\tag{21}
$$

Obviously, the tangency condition of MM algorithm is satisfied: $M(\hat{\theta}_{ij} \mid \theta_{ij}^{(t)}) = O(\hat{\theta}_{ij} \mid \theta_{ij}^{(t)}) = D(x_{ij} \mid \theta_{ij}^{(t)})$, and the control condition is also satisfied for all $\theta$: $M(\hat{\theta}_{ij} \mid \theta_{ij}^{(t)}) \geq O(\hat{\theta}_{ij} \mid \theta_{ij}^{(t)})$ because of Equation (20), similar to the case in traditional generalized PCA (Landgraf & Lee, 2020).

Then summing $M(\hat{\theta}_{ij} \mid \theta_{ij}^{(t)})$ to get $M(\hat{\Theta} \mid \Theta^{(t)})$:

$$
\begin{aligned}
M(\hat{\Theta} \mid \Theta^{(t)}) &= \sum_{i,j} M(\hat{\theta}_{ij} \mid \theta_{ij}^{(t)}) \\
&= \sum_{i,j} v_i^{(t)}\left[\hat{\theta}_{ij} - \left(\theta_{ij}^{(t)} + \frac{w_{ij}}{v_i^{(t)}}\left(\frac{1}{\theta_{ij}^{(t)}} + x_{ij}\right)\right)\right]^2 + C.
\end{aligned}
\tag{22}
$$

Further, we introduce the following substitution:

$$
z_{ij}^{(t)} = \theta_{ij}^{(t)} + \frac{w_{ij}}{v_i^{(t)}}\left(\frac{1}{\theta_{ij}^{(t)}} + x_{ij}\right).
\tag{23}
$$

Equation (22) can be rewritten as:

$$
\begin{aligned}
M(\hat{\Theta} \mid \Theta^{(t)}) &= \sum_{i,j} v_i^{(t)}\left[\hat{\theta}_{ij} - z_{ij}^{(t)}\right]^2 + C \\
&= \sum_{i,j} v_i^{(t)}\left[\left(\hat{\theta}_{ij}\right)^2 \frac{1}{\hat{\theta}_{ij}} - z_{ij}^{(t)}\right]^2 + C \\
&\approx \sum_{i,j} v_i^{(t)}\left[\left(\theta_{ij}^{(t)}\right)^2 \frac{1}{\hat{\theta}_{ij}} - z_{ij}^{(t)}\right]^2 + C.
\end{aligned}
\tag{24}
$$

In $k$-dimensional subspace, the estimation of $\hat{\theta}_{ij}$ is defined in Equation (9), so $M(\hat{\Theta} \mid \Theta^{(t)})$ can be rewritten as

$$
\begin{aligned}
&\sum_{i,j} v_i^{(t)}\left[(\theta_{ij}^{(t)})^2\left\{\left[\mathbf{U}\mathbf{U}^T\left(\tilde{\boldsymbol{\eta}}_i(\theta) + \boldsymbol{\mu}\right)\right]_j - \mu_j\right\} - z_{ij}^{(t)}\right]^2 + C \\
&= \sum_{i,j} v_i^{(t)}(\theta_{ij}^{(t)})^4\left[\left[\mathbf{U}\mathbf{U}^T\left(\tilde{\boldsymbol{\eta}}_i(\theta) + \boldsymbol{\mu}\right)\right]_j - \mu_j - \frac{z_{ij}^{(t)}}{(\theta_{ij}^{(t)})^2}\right]^2 + C \\
&\leq \sum_{i,j} q_{ij}^{(t)}\left[\left[\mathbf{U}\mathbf{U}^T\left(\tilde{\boldsymbol{\eta}}_i(\theta) + \boldsymbol{\mu}\right)\right]_j - \left(p_{ij}^{(t)} + \mu_j\right)\right]^2 + C.
\end{aligned}
\tag{25}
$$

In the above equation, we introduce the following expressions for ease of notation:

$$
q_i^{(t)} := v_i^{(t)}\max_j[(\theta_{ij}^{(t)})^4], \; p_{ij}^{(t)} := z_{ij}^{(t)}/(\theta_{ij}^{(t)})^2.
\tag{26}
$$

Let $\mathbf{P}^{(t)} = [p_{ij}^{(t)}]$, $\mathbf{H} = [\tilde{\boldsymbol{\eta}}_1(\theta), \tilde{\boldsymbol{\eta}}_2(\theta), \cdots, \tilde{\boldsymbol{\eta}}_n(\theta)]^T$ and $q_i^{(t)}$ is the diagonal element of the diagonal matrix $\mathbf{Q}^{(t)}$. $M(\Theta \mid \Theta^{(t)})$ can be further written as the matrix form:

$$
M = \left\|\begin{array}{l}\left(\mathbf{Q}^{(t)}\right)^{\frac{1}{2}}\left(\mathbf{H} + \mathbf{1}_n(\boldsymbol{\mu}^{(t)})^T\right)\mathbf{U}^{(t)}\left(\mathbf{U}^{(t)}\right)^T \\ -\left(\mathbf{Q}^{(t)}\right)^{\frac{1}{2}}\left(\mathbf{P}^{(t)} + \mathbf{1}_n(\boldsymbol{\mu}^{(t)})^T\right)\end{array}\right\|_F^2 + C,
\tag{27}
$$

where $\|\cdot\|_F$ is the Frobenius Norm. We can minimize $M$ by iteration to find the projection matrix $\mathbf{U}$ and the mean vector $\boldsymbol{\mu}$:

$$\arg\min_{\mathbf{U}^T\mathbf{U}=\mathbf{I}_k} \left\| \begin{array}{c} \left(\mathbf{Q}^{(t)}\right)^{\frac{1}{2}} \left(\mathbf{H}+\mathbf{1}_n(\boldsymbol{\mu}^{(t)})^T\right) \mathbf{U}^{(t)} \left(\mathbf{U}^{(t)}\right)^T \\ -\left(\mathbf{Q}^{(t)}\right)^{\frac{1}{2}} \left(\mathbf{P}^{(t)}+\mathbf{1}_n(\boldsymbol{\mu}^{(t)})^T\right) \end{array} \right\|_F^2 + \mathbf{C}. \tag{28}$$

### B.2. The modification of U and $\boldsymbol{\mu}$

Holding $\boldsymbol{\mu}$ fixed, the minimum of Equation (28) can be found by updating $\mathbf{U}$. According to the property of Frobenius Norm, it can be rewritten in the form of a trace of matrices. Therefore, Equation (27) can be rewritten as follows:

$$
\begin{aligned}
M(\boldsymbol{\Theta}\,|\,\boldsymbol{\Theta}^{(t)}) &= \left\|(\mathbf{Q}^{(t)})^{1/2}\left(\mathbf{H}+\mathbf{1}_n(\boldsymbol{\mu}^{(t)})^T\right)\mathbf{U}^{(t)}(\mathbf{U}^{(t)})^T - (\mathbf{Q}^{(t)})^{1/2}\left(\mathbf{P}^{(t)}+\mathbf{1}_n(\boldsymbol{\mu}^{(t)})^T\right)\right\|_F^2 \\
&= \mathrm{tr}\left[\mathbf{U}^{(t)}(\mathbf{U}^{(t)})^T\left(\mathbf{H}+\mathbf{1}_n(\boldsymbol{\mu}^{(t)})^T\right)^T\left((\mathbf{Q}^{(t)})^{1/2}\right)^T(\mathbf{Q}^{(t)})^{1/2}\left(\mathbf{H}+\mathbf{1}_n(\boldsymbol{\mu}^{(t)})^T\right)\mathbf{U}^{(t)}(\mathbf{U}^{(t)})^T\right] \\
&\quad - \mathrm{tr}\left[\mathbf{U}^{(t)}(\mathbf{U}^{(t)})^T\left(\mathbf{H}+\mathbf{1}_n(\boldsymbol{\mu}^{(t)})^T\right)^T\left((\mathbf{Q}^{(t)})^{1/2}\right)^T(\mathbf{Q}^{(t)})^{1/2}\left(\mathbf{P}^{(t)}+\mathbf{1}_n(\boldsymbol{\mu}^{(t)})^T\right)\right] \\
&\quad - \mathrm{tr}\left[\left(\mathbf{P}^{(t)}+\mathbf{1}_n(\boldsymbol{\mu}^{(t)})^T\right)\left((\mathbf{Q}^{(t)})^{1/2}\right)^T(\mathbf{Q}^{(t)})^{1/2}\left(\mathbf{H}+\mathbf{1}_n(\boldsymbol{\mu}^{(t)})^T\right)^T\mathbf{U}^{(t)}(\mathbf{U}^{(t)})^T\right] \\
&= \mathrm{tr}\Bigg[(\mathbf{U}^{(t)})^T\Big\{\left(\mathbf{H}+\mathbf{1}_n(\boldsymbol{\mu}^{(t)})^T\right)^T\mathbf{Q}^{(t)}\left(\mathbf{P}^{(t)}+\mathbf{1}_n(\boldsymbol{\mu}^{(t)})^T\right) \\
&\qquad\qquad + \left(\mathbf{P}^{(t)}+\mathbf{1}_n(\boldsymbol{\mu}^{(t)})^T\right)^T\mathbf{Q}^{(t)}\left(\mathbf{H}+\mathbf{1}_n(\boldsymbol{\mu}^{(t)})^T\right) \\
&\qquad\qquad - \left(\mathbf{H}+\mathbf{1}_n(\boldsymbol{\mu}^{(t)})^T\right)^T\mathbf{Q}^{(t)}\left(\mathbf{H}+\mathbf{1}_n(\boldsymbol{\mu}^{(t)})^T\right)\Big\}\mathbf{U}^{(t)}\Bigg]
\end{aligned}
\tag{29}
$$

Denote that:

$$
\begin{aligned}
\mathbf{F}^{(t)}\left(\mathbf{H},\boldsymbol{\mu}^{(t)},\mathbf{P}^{(t)},\mathbf{Q}^{(t)}\right) &= \left(\mathbf{H}+\mathbf{1}_n(\boldsymbol{\mu}^{(t)})^T\right)^T\mathbf{Q}^{(t)}\left(\mathbf{P}^{(t)}+\mathbf{1}_n(\boldsymbol{\mu}^{(t)})^T\right) \\
&\quad + \left(\mathbf{P}^{(t)}+\mathbf{1}_n(\boldsymbol{\mu}^{(t)})^T\right)^T\mathbf{Q}^{(t)}\left(\mathbf{H}+\mathbf{1}_n(\boldsymbol{\mu}^{(t)})^T\right) \\
&\quad - \left(\mathbf{H}+\mathbf{1}_n(\boldsymbol{\mu}^{(t)})^T\right)^T\mathbf{Q}^{(t)}\left(\mathbf{H}+\mathbf{1}_n(\boldsymbol{\mu}^{(t)})^T\right).
\end{aligned}
\tag{30}
$$

The minimization of $M(\boldsymbol{\Theta}\,|\,\boldsymbol{\Theta}^{(t)})$ is equivalent to:

$$\arg\max M(\boldsymbol{\Theta}\,|\,\boldsymbol{\Theta}^{(t)}) = \arg\max_{(\mathbf{U})^T\mathbf{U}=\mathbf{I}_k} \mathrm{tr}\left[(\mathbf{U}^{(t)})^T\mathbf{F}^{(t)}(\mathbf{H},\boldsymbol{\mu}^{(t)},\mathbf{P}^{(t)},\mathbf{Q}^{(t)})\mathbf{U}^{(t)}\right], \tag{31}$$

where the trace is maximized with respect to orthonormal $\mathbf{U}$ by the first $k$ eigenvectors of $\mathbf{F}^{(t)}(\mathbf{H},\boldsymbol{\mu}^{(t)},\mathbf{P}^{(t)},\mathbf{Q}^{(t)})$.

On the other hand, holding $\mathbf{U}$ fixed, the optimal $\boldsymbol{\mu}$ can be found by minimizing $M(\boldsymbol{\Theta}\,|\,\boldsymbol{\Theta}^{(t)})$:

$$
\begin{aligned}
M(\boldsymbol{\Theta}\,|\,\boldsymbol{\Theta}^{(t)}) &= \left\|(\mathbf{Q}^{(t)})^{1/2}\left(\mathbf{H}+\mathbf{1}_n(\boldsymbol{\mu}^{(t)})^T\right)\mathbf{U}^{(t)}(\mathbf{U}^{(t)})^T - (\mathbf{Q}^{(t)})^{1/2}\left(\mathbf{P}^{(t)}+\mathbf{1}_n(\boldsymbol{\mu}^{(t)})^T\right)\right\|_F^2 \\
&= \left\|(\mathbf{Q}^{(t)})^{1/2}\mathbf{H}\mathbf{U}^{(t)}(\mathbf{U}^{(t)})^T + (\mathbf{Q}^{(t)})^{1/2}\mathbf{1}_n(\boldsymbol{\mu}^{(t)})^T\mathbf{U}^{(t)}(\mathbf{U}^{(t)})^T - (\mathbf{Q}^{(t)})^{1/2}\mathbf{P}^{(t)} - (\mathbf{Q}^{(t)})^{1/2}\mathbf{1}_n(\boldsymbol{\mu}^{(t)})^T\right\|_F^2 \\
&= \left\|(\mathbf{Q}^{(t)})^{1/2}\left(\mathbf{H}\mathbf{U}^{(t)}(\mathbf{U}^{(t)})^T - \mathbf{P}^{(t)}\right) + (\mathbf{Q}^{(t)})^{1/2}\mathbf{1}_n(\boldsymbol{\mu}^{(t)})^T\left(\mathbf{U}^{(t)}(\mathbf{U}^{(t)})^T - \mathbf{I}_d\right)\right\|_F^2.
\end{aligned}
\tag{32}
$$

Let $\mathbf{A}^{(t)} := \left(\mathbf{H}\mathbf{U}^{(t)}(\mathbf{U}^{(t)})^T - \mathbf{P}^{(t)}\right) \in \mathbb{R}^{n\times d}$ and $\mathbf{B}^{(t)} := \left(\mathbf{U}^{(t)}(\mathbf{U}^{(t)})^T - \mathbf{I}_d\right) \in \mathbb{R}^{d\times d}$, and we can convert the computation

of Frobenius Norm to the computation of trace of matrices:

$$
\begin{aligned}
M(\mathbf{\Theta}\,|\,\mathbf{\Theta}^{(t)}) =\ & \mathrm{tr}\left\{\left[(\mathbf{Q}^{(t)})^{1/2}\mathbf{A}^{(t)} + (\mathbf{Q}^{(t)})^{1/2}\mathbf{1}_n(\boldsymbol{\mu}^{(t)})^T\mathbf{B}^{(t)}\right]^T \left[(\mathbf{Q}^{(t)})^{1/2}\mathbf{A}^{(t)} + (\mathbf{Q}^{(t)})^{1/2}\mathbf{1}_n(\boldsymbol{\mu}^{(t)})^T\mathbf{B}^{(t)}\right]\right\} \\
=\ & \mathrm{tr}\left\{(\mathbf{A}^{(t)})^T\left((\mathbf{Q}^{(t)})^{1/2}\right)^T(\mathbf{Q}^{(t)})^{1/2}\mathbf{1}_n(\boldsymbol{\mu}^{(t)})^T\mathbf{B}^{(t)}\right\} \\
& + \mathrm{tr}\left\{(\mathbf{B}^{(t)})^T\boldsymbol{\mu}^{(t)}(\mathbf{1}_n)^T\left((\mathbf{Q}^{(t)})^{1/2}\right)^T(\mathbf{Q}^{(t)})^{1/2}\mathbf{A}^{(t)}\right\} \\
& + \mathrm{tr}\left\{(\mathbf{B}^{(t)})^T\boldsymbol{\mu}^{(t)}(\mathbf{1}_n)^T\left((\mathbf{Q}^{(t)})^{1/2}\right)^T(\mathbf{Q}^{(t)})^{1/2}\mathbf{1}_n(\boldsymbol{\mu}^{(t)})^T\mathbf{B}^{(t)}\right\} \\
=\ & \mathrm{tr}\left\{(\mathbf{A}^{(t)})^T\mathbf{Q}^{(t)}\mathbf{1}_n(\boldsymbol{\mu}^{(t)})^T\mathbf{B}^{(t)}\right\} + \mathrm{tr}\left\{(\mathbf{B}^{(t)})^T\boldsymbol{\mu}^{(t)}(\mathbf{1}_n)^T\mathbf{Q}^{(t)}\mathbf{A}^{(t)}\right\} \\
& + \mathrm{tr}\left\{(\mathbf{B}^{(t)})^T\boldsymbol{\mu}^{(t)}(\mathbf{1}_n)^T\mathbf{Q}^{(t)}\mathbf{1}_n(\boldsymbol{\mu}^{(t)})^T\mathbf{B}^{(t)}\right\} \\
=\ & 2\,\mathrm{tr}\left\{(\mathbf{A}^{(t)})^T\mathbf{Q}^{(t)}\mathbf{1}_n(\boldsymbol{\mu}^{(t)})^T\mathbf{B}^{(t)}\right\} + \mathrm{tr}\left\{(\mathbf{B}^{(t)})^T(\boldsymbol{\mu}^{(t)})(\mathbf{1}_n)^T\mathbf{Q}^{(t)}\mathbf{1}_n(\boldsymbol{\mu}^{(t)})^T\mathbf{B}^{(t)}\right\}.
\end{aligned}
\tag{33}
$$

Then we take the partial derivative of two parts of $M(\mathbf{\Theta}\,|\,\mathbf{\Theta}^{(t)})$ with respect to $\boldsymbol{\mu}^{(t)}$. The partial derivative of former part in Equation (33) is as follows:

$$
\begin{aligned}
& \frac{\partial}{\partial\boldsymbol{\mu}^{(t)}}\,\mathrm{tr}\left\{(\mathbf{A}^{(t)})^T\mathbf{Q}^{(t)}\mathbf{1}_n(\boldsymbol{\mu}^{(t)})^T\mathbf{B}^{(t)}\right\} \\
=\ & \frac{\partial}{\partial\boldsymbol{\mu}^{(t)}}\,\mathrm{tr}\left\{\mathbf{B}^{(t)}(\mathbf{A}^{(t)})^T\mathbf{Q}^{(t)}\mathbf{1}_n(\boldsymbol{\mu}^{(t)})^T\right\} \\
=\ & \frac{\partial}{\partial\boldsymbol{\mu}^{(t)}}\,\mathrm{tr}\left\{\boldsymbol{\mu}^{(t)}(\mathbf{1}_n)^T(\mathbf{Q}^{(t)})^T\mathbf{A}^{(t)}(\mathbf{B}^{(t)})^T\right\} \\
=\ & \left((\mathbf{1}_n)^T(\mathbf{Q}^{(t)})^T\mathbf{A}^{(t)}(\mathbf{B}^{(t)})^T\right)^T \\
=\ & \mathbf{B}^{(t)}(\mathbf{A}^{(t)})^T\mathbf{Q}^{(t)}\mathbf{1}_n.
\end{aligned}
\tag{34}
$$

As for the partial derivative of latter part in Equation (33), because $(\mathbf{1}_n)^T\mathbf{Q}^{(t)}\mathbf{1}_n$ is essentially the sum of all the elements of $\mathbf{Q}^{(t)}$, the partial derivative of this part can be expressed as:

$$
\begin{aligned}
& \frac{\partial}{\partial\boldsymbol{\mu}^{(t)}}\,\mathrm{tr}\left\{(\mathbf{B}^{(t)})^T\boldsymbol{\mu}^{(t)}(\mathbf{1}_n)^T\mathbf{Q}^{(t)}\mathbf{1}_n(\boldsymbol{\mu}^{(t)})^T\mathbf{B}^{(t)}\right\} \\
=\ & (\mathbf{1}_n)^T\mathbf{Q}^{(t)}\mathbf{1}_n\frac{\partial}{\partial\boldsymbol{\mu}^{(t)}}\,\mathrm{tr}\left\{(\mathbf{B}^{(t)})^T\boldsymbol{\mu}^{(t)}(\boldsymbol{\mu}^{(t)})^T\mathbf{B}^{(t)}\right\} \\
=\ & (\mathbf{1}_n)^T\mathbf{Q}^{(t)}\mathbf{1}_n\frac{\partial}{\partial\boldsymbol{\mu}^{(t)}}\,\mathrm{tr}\left\{(\boldsymbol{\mu}^{(t)})^T\mathbf{B}^{(t)}(\mathbf{B}^{(t)})^T\boldsymbol{\mu}^{(t)}\right\} \\
=\ & 2\,(\mathbf{1}_n)^T\mathbf{Q}^{(t)}\mathbf{1}_n\mathbf{B}^{(t)}(\mathbf{B}^{(t)})^T\boldsymbol{\mu}^{(t)}.
\end{aligned}
\tag{35}
$$

Therefore, we can get:

$$
\frac{\partial M(\mathbf{\Theta}\,|\,\mathbf{\Theta}^{(t)})}{\partial\boldsymbol{\mu}^{(t)}} = 2\mathbf{B}^{(t)}(\mathbf{A}^{(t)})^T\mathbf{Q}^{(t)}\mathbf{1}_n + 2\,(\mathbf{1}_n)^T\mathbf{Q}^{(t)}\mathbf{1}_n\mathbf{B}^{(t)}(\mathbf{B}^{(t)})^T\boldsymbol{\mu}^{(t)}.
\tag{36}
$$

Then let Equation (36) equals to 0 to find its extreme point:

$$
\mathbf{B}^{(t)}(\mathbf{B}^{(t)})^T\boldsymbol{\mu}^{(t)} = -\left((\mathbf{1}_n)^T\mathbf{Q}^{(t)}\mathbf{1}_n\right)^{-1}\mathbf{B}^{(t)}(\mathbf{A}^{(t)})^T\mathbf{Q}^{(t)}\mathbf{1}_n.
\tag{37}
$$

In the above equation, $\mathbf{B}^{(t)}(\mathbf{B}^{(t)})^T$ can be simplified:

$$
\begin{aligned}
&\mathbf{B}^{(t)}(\mathbf{B}^{(t)})^T \\
&= \left(\mathbf{I}_d - \mathbf{U}^{(t)}(\mathbf{U}^{(t)})^T\right)\left(\mathbf{I}_d - \mathbf{U}^{(t)}(\mathbf{U}^{(t)})^T\right) \\
&= \mathbf{U}^{(t)}(\mathbf{U}^{(t)})^T\mathbf{U}^{(t)}(\mathbf{U}^{(t)})^T - 2\mathbf{U}^{(t)}(\mathbf{U}^{(t)})^T + \mathbf{I}_d \\
&= \mathbf{I}_d - \mathbf{U}^{(t)}(\mathbf{U}^{(t)})^T \\
&= -\mathbf{B}^{(t)}.
\end{aligned}
\tag{38}
$$

Further substitute $\mathbf{A}^{(t)} := \left(\mathbf{H}\mathbf{U}^{(t)}(\mathbf{U}^{(t)})^T - \mathbf{P}^{(t)}\right)$ and $\mathbf{B}^{(t)} := \left(\mathbf{U}^{(t)}(\mathbf{U}^{(t)})^T - \mathbf{I}_d\right)$ into Equation (37), and $\boldsymbol{\mu}^{(t)}$ is finally derived as follows:

$$
\begin{aligned}
-\mathbf{B}^{(t)}\boldsymbol{\mu}^{(t)} &= -\left((\mathbf{1}_n)^T\mathbf{Q}^{(t)}\mathbf{1}_n\right)^{-1}\mathbf{B}^{(t)}(\mathbf{A}^{(t)})^T\mathbf{Q}^{(t)}\mathbf{1}_n \\
\boldsymbol{\mu}^{(t)} &= \left((\mathbf{1}_n)^T\mathbf{Q}^{(t)}\mathbf{1}_n\right)^{-1}\left(\mathbf{H}\mathbf{U}^{(t)}(\mathbf{U}^{(t)})^T - \mathbf{P}^{(t)}\right)^T\mathbf{Q}^{(t)}\mathbf{1}_n.
\end{aligned}
\tag{39}
$$

## C. Detailed Composition of Experimental Dataset.

### C.1. Description of MSTAR dataset

There are three imaging depression angles in MSTAR: 15°, 17° and 30°, each depression consists of data from complete azimuth angles ( 0°-360°). MSTAR has divided all images into four quadrants according to their azimuth angles, 0°-90°, 90°-180°, 180°-270°, 270°-360°, respectively. While dataset at depression 15° and 17° both include 10 categories: BMP2, BTR70, T72, 2S1, BRDM2, D7, BTR60, T62, ZIL131, ZSU234, dataset at depression 30° only includes 3 categories: 2S1, BRDM2 and ZSU34.

Table 5: Description of MSTAR dataset.

| Depression | 17° | | | | 15° | 30° |
|---|---|---|---|---|---|---|
| Azimuth | 0°-90° | 90°-180° | 180°-270° | 270°-360° | 0°-360° | 0°-360° |
| 2S1 | 71 | 74 | 79 | 75 | 274 | 288 |
| BMP2 | 61 | 60 | 51 | 61 | 195 | N/A |
| BRDM2 | 72 | 73 | 80 | 73 | 274 | 420 |
| BTR60 | 68 | 68 | 60 | 60 | 195 | N/A |
| BTR70 | 56 | 52 | 61 | 64 | 196 | N/A |
| D7 | 70 | 82 | 68 | 79 | 274 | N/A |
| T62 | 73 | 74 | 80 | 72 | 273 | N/A |
| T72 | 62 | 56 | 51 | 63 | 196 | N/A |
| ZIL131 | 72 | 74 | 80 | 73 | 274 | N/A |
| ZSU234 | 72 | 73 | 80 | 74 | 274 | 406 |

### C.2. Description of SAR-AIRcraft-1.0 dataset

The images in this dataset are obtained from the satellite Gaofen-3 with single polarization, a spatial resolution of 1 meter and Spotlight imaging mode. This dataset contains 4,368 images and covers 7 aircraft categories, namely A220, A320/321, A330, ARJ21, Boeing737, Boeing787 and other, where "other" represents aircraft instances that do not belong to the mentioned six categories.

Table 6: Description of SAR-AIRcraft-1.0 dataset.

| Class | A220 | A320/321 | A330 | ARJ21 | Boeing737 | Boeing787 | other | all |
|---|---|---|---|---|---|---|---|---|
| Number | 2065 | 939 | 290 | 713 | 1495 | 1677 | 2041 | 9220 |

