# OpenReview forum: "Gamma Distribution PCA-Enhanced Feature Learning for Angle-Robust SAR Target Recognition"
_ICML.cc/2025/Conference — ICML 2025 poster_

### Official Review · Reviewer_Ub2X · 2025-03-14

**Overall Recommendation:** 2

**Summary:**

This paper proposes a Gamma-Distribution Principal Component Analysis (ΓPCA) method for angle-robust SAR target recognition. The key idea is to integrate the Gamma distribution into PCA to account for the statistical properties of SAR data, thereby enhancing feature extraction across varying observation angles. The authors claim that ΓPCA-derived convolution kernels can capture angle-invariant features without adding additional computational burden. The method is evaluated on the MSTAR dataset with ResNet and ViT backbones.

**Claims And Evidence:**

no

**Essential References Not Discussed:**

no

**Ethical Review Flag:**

Flag this paper for an ethics review.

**Experimental Designs Or Analyses:**

The paper compares against ViT, ResNet, and some recent models but ignores SAR-specific methods such as UIU-Net or ASC-based approaches.
 The study shows ΓPCA with different architectures but does not isolate the contributions of different components (e.g., Gamma assumption vs. PCA).
The study does not explore the effect of different kernel sizes, training strategies, or dataset variations.
Real-World Relevance: No analysis is provided on how the method performs under different noise conditions, sensor types, or imaging artifacts.

**Methods And Evaluation Criteria:**

The study is based entirely on MSTAR, which, while widely used, is a relatively small dataset that may not reflect real-world SAR conditions.
 The authors compare ΓPCA with ViT, ResNet, and a few SOTA models, but crucial SAR-specific baselines (e.g., UIU-Net, ASC-based models) are missing.
No statistical significance tests (e.g., t-tests, Wilcoxon tests) are performed to validate improvements.

**Other Comments Or Suggestions:**

no more

**Other Strengths And Weaknesses:**

no more.

**Questions For Authors:**

no more

**Relation To Broader Scientific Literature:**

The paper references many relevant SAR and deep learning studies but fails to discuss critical related work in:
Statistical modeling of SAR images (Weibull, K-distribution, etc.)
Deep learning for SAR ATR (UIU-Net, few-shot SAR methods)
Angle-invariant feature learning in computer vision

**Theoretical Claims:**

The paper extends PCA to a Gamma-distributed setting, claiming it better represents SAR image statistics.
The derivation of ΓPCA is mathematically rigorous, but no theoretical proof is provided that ΓPCA is better than traditional PCA or other low-rank approximations.
The study assumes that angle variation can be mitigated by low-rank feature extraction, but does not establish a formal theoretical link between angle-robustness and Gamma-PCA projections.

---

> ### Author Rebuttal · Authors · 2025-04-01
>
> We sincerely appreciate the reviewer’s valuable comments. We will enhance the related work and include more experimental results in our final version. Below are our detailed responses to each point.
> > Q1. MSTAR is small.
>
> **A1:** We constructed a new dataset based on SAR-AIRcraft-1.0. A detailed description of the dataset and the experimental results can be found in **Answer1** for **Reviewer Otus**.
>
> > Q2. SAR-specific baselines.
>
> **A2:** We incorporat the ASC-based method MSNet-PIHA [R10] into our experiments. The comparison results are shown below.
>
> |Methods|A-R Test|AD-R Test| Overall Acc. (%)|
> |---|---|---|---|
> |MSNet-PIHA| 68.08±0.03|55.75±0.06| 64.20±0.03|
> |ΓPCA-Resnet|73.79±0.02|52.96±0.05|67.23±0.01|
> |ΓPCA-ViT|78.48±0.01|70.67±0.16|76.02±0.01|
>
> MSNet-PIHA performs well when trained on diverse azimuth angles [R10] but struggles with limited-angle data, which can lead to overfitting and suboptimal predictions. While ASC information enhances feature extraction with limited data, severe azimuth constraints remain challenging. In contrast, ΓPCA demonstrates better angle robustness.
>
> [R10] Huang, Z., et al. Physics inspired hybrid attention for SAR target recognition. *ISPRS*, 2024.
>
> > Q3. Statistical significance tests.
>
> **A3:** We conduct significance tests by repeating all experiments in our paper three times and reported the mean and variance to reduce the influence of randomness. Part of the updated experimental results are presented below.
>
> Table R1. Partially updated recognition results of Table 1.
>
> |Methods|A-R Test|AD-R Test|Overall Acc. (%)|
> |---|---|---|---|
> |Resnet|72.57±0.25|43.95±0.01|63.62±0.14|
> |ViT|78.12±0.04|60.11±0.24|72.45±0.04|
> |ΓPCA-Resnet|73.79±0.02|52.96±0.05|67.23±0.01|
> |ΓPCA-ViT|78.48±0.01|70.67±0.16|76.02±0.01|
>
> More results can be found at: https://anonymous.4open.science/r/ICML2025-E1F8.
>
> > Q4. Theoretical proof of ΓPCA is better than PCA.
>
> **A4:** Previous work in [R11] has proven that generalized PCA based on exponential family distributions theoretically outperforms traditional PCA. As a variant, our ΓPCA, based on the Gamma distribution, inherits this theoretical advantage.
>
> [R11] Landgraf, A. J., et al. Generalized principal component analysis: Projection of saturated model parameters. *Technometrics*, 2020.
>
> > Q5. Theoretical link between angle-robustness and ΓPCA projections.
>
> **A5:** Under different orientation observation angles, the statistics of the same class of SAR targets are similar. Base on this characteristic, ΓPCA assumes that the SAR target image follows the Gamma distribution, and then derives ΓPCA projection which can extract the principal component information, i.e., the low-rank information, of these SAR targets. Then the projection matrix is used to construct the convolution kernel to extract the SAR target features, which are characterized by a low sensitivity to angle.
> > Q6. Comparison of different components.
>
> **A6:** We provide an isolated comparison of different components (see Table 3, page 8, PCA-ViT vs. ΓPCA-ViT). As demonstrated in Table 3, ΓPCA consistently outperforms conventional PCA with ViT backbone for SAR data. For example, with a kernel size of 17, PCA-ViT and ΓPCA-ViT achieve accuracies of 73.52\% and 76.01\%, respectively, demonstrating its effectiveness in non-Gaussian SAR data analysis.
> > Q7. Effects of different settings.
>
> **A7:** Kernel size analysis is presented in Sec. 4.3 (Page 8, Left Column, Lines 410–434), with quantitative comparisons in Table 3. To isolate the impact of ΓPCA, all experiments exploit the same training settings (optimizer and learning rate: the default configurations of the original backbones, batch size: 32), ensuring a fair comparison within the scope of the paper on fundamental ΓPCA theory for SAR data.
> > Q8. Real-World Relevance.
>
> **A8:** We conducted experiments under various noise conditions, focusing on multiplicative noise, i.e. speckle, the most common noise type in SAR image. Noise levels are quantified by the Equivalent Number of Looks (ENL), defined as: μ^2/σ^2, where μ and σ are mean and standard deviation of pixel values. A higher ENL value indicates lower noise.
>
> |ENL|ViT|ΓPCA+ViT|
> |---|---|---|
> |Noiseless|70.61|76.01|
> |20|51.51|69.60|
> |10|47.22|62.19|
> |5|45.30|56.23|
> |2|41.20|48.09|
>
> More results are available at: https://anonymous.4open.science/r/ICML2025-E1F8. These results show the anti-noise performance of ΓPCA, attributed to its ability to extract principal components while filtering out noise-dominated ones.
>
> Our study primarily addresses the challenge of angle-induced performance degradation in SAR ATR. Extensive experiments validate the dual robustness of ΓPCA to angular variations and speckle noise. While sensor types and imaging artifacts are also important for SAR ATR, these aspects fall beyond the scope of our current work, as they pose distinct research challenges. We will further clarify the influence of these two aspects in the final version.

---

> > ### Comment · Reviewer_Ub2X · 2025-04-07
> >
> > Despite acknowledging the authors’ extensive efforts to address reviewer concerns, several major issues remain unresolved. In particular:
> >
> > Justification of the Gamma Distribution:
> > The authors do not convincingly establish the theoretical link between the Gamma distribution assumption and the underlying SAR imaging physics. Why is the Gamma distribution particularly well-suited for modeling SAR images, and how does this choice offer clear advantages over traditional or even more modern methods? The explanation provided is vague and does not substantiate the claim that this approach is better tailored to the specific noise characteristics and imaging artifacts in SAR data.
> >
> > Reinventing an Established Method:
> > PCA is a long-established technique and the improvement via a Gamma-distribution-based variant is, at best, incremental. The authors have not provided compelling evidence that the proposed ΓPCA adds sufficient value over conventional PCA or other advanced dimensionality reduction methods. Given that numerous state-of-the-art approaches with robust theoretical foundations demonstrate superior performance, the reliance on a modified PCA framework appears to be a regression rather than an advancement.
> >
> > Comparative Performance with Recent State-of-the-Art:
> > Although the rebuttal includes additional experimental results, the reported accuracies still do not convincingly outperform many recent methods. There is a lack of discussion regarding why a fundamental method like PCA, even in a generalized Gamma form, should be preferred over contemporary approaches that often incorporate more sophisticated theoretical developments and yield higher recognition performance.
> >
> > About angle information in SAR recognition:
> > The focus on angle-induced variations in SAR image recognition appears overstated. While the manuscript emphasizes the importance of angle robustness, identifying SAR images fundamentally falls under the broader umbrella of image recognition, where standard data augmentation techniques can effectively mitigate variations introduced by different viewing angles. The authors have not provided compelling evidence that angle variations are a uniquely critical challenge for SAR recognition that cannot be addressed by simpler and more proven methods. Furthermore, the paper does not include any experimental validation to compare the proposed ΓPCA method against data augmentation baselines. Without demonstrating that simple augmentation strategies fail to yield comparable or superior results, the motivation for developing a Gamma-distribution-based variant of PCA becomes questionable, especially since PCA is a well-established method and its modification in this context appears incremental.
> >
> > Overall, while the authors have made a considerable effort to address specific reviewer points, the manuscript does not sufficiently justify the theoretical and practical merits of ΓPCA within the context of SAR target recognition. Given these unresolved issues, especially the unclear motivation for adopting the Gamma distribution and the reliance on an aging methodology in the face of modern alternatives, I remain inclined to recommend rejection.

---

### Official Review · Reviewer_otus · 2025-03-14

**Overall Recommendation:** 4

**Summary:**

The paper proposes a Gamma-Distribution Principal Component Analysis ($\Gamma$PCA) model to enhance the robustness of Synthetic Aperture Radar (SAR) target recognition against angle variations. The key idea is to leverage the Gamma distribution to extract low-rank features that are invariant to changes in azimuth and depression angles. The authors derive a consistency projection matrix from $\Gamma$PCA to construct convolutional kernels that capture angle-insensitive information. The proposed method is integrated into deep learning backbones (ResNet and ViT) and validated on the MSTAR dataset, demonstrating improved robustness and performance compared to baseline models.

**Claims And Evidence:**

* The authors take the non-Gaussian nature of SAR into consideration and deveplot the $\Gamma$PCA model accordingly.
* The experimental results on the MSTAR dataset demonstrate that $\Gamma$PCA can improve the robustness of ResNet and ViT models against angle variations.

**Essential References Not Discussed:**

Based on the information provided, there are no obvious essential references missing.

**Experimental Designs Or Analyses:**

* The authors conduct experiments on the MSTAR dataset to evaluate the azimuth robustness and azimuth & depression robustness of the proposed method.
* Experiments with state-of-the-art models (e.g., ViT and ResNet) further validate the effectiveness of the proposed method.

**Methods And Evaluation Criteria:**

The evaluation criteria look good.

**Other Comments Or Suggestions:**

* Line 143, the formulation lacks ending period.
* Figure 4 can be lifted to former pages.
* Figures 5 and 6, the same label repeats 6 times, please simplify it. Also, the font can be larger.

**Other Strengths And Weaknesses:**

Strengths:
* The paper presents a novel approach to addressing a significant challenge in SAR target recognition: robustness to angle variations.
* The method is demonstrated to be effective across different deep learning architectures (ResNet and ViT), highlighting its generalizability.

Weaknesses:
* The experimental validation is limited to a single dataset (MSTAR), which may restrict the generalizability of the findings.

**Questions For Authors:**

There is no further questions for the authors.

**Relation To Broader Scientific Literature:**

The key contributions are not related to the broader scientific literature.

**Theoretical Claims:**

There is no proof for theoretical claims to be checked.

---

> ### Author Rebuttal · Authors · 2025-04-01
>
> > Comment1. The experimental validation is limited to a single dataset (MSTAR).
>
> **Answer1:**
> Thanks for pointing out this important concern. We acknowledge this limitation. Unlike optical imagery, SAR data is challenging to obtain at scale due to sensor costs, operational constraints, and the factor of security sensitivities. Therefore, most studies [R3-R6] rely solely on MSTAR for evaluation. To enhance experimental validation, we construct a dataset based on SAR-AIRcraft-1.0, a widely used public SAR aircraft detection dataset. Compared to MSTAR, SAR-AIRcraft-1.0 offers richer data diversity and more complex operational scenarios. We extract individual targets from the original imagery to create a dedicated SAR ATR dataset. Below, we provide a detailed description of its composition.
>
> | Class  | A220 | A320/321 | A330 | ARJ21 | Boeing737 | Boeing787 | other | all  |
> | ------ | ---- | -------- | ---- | ----- | --------- | --------- | ----- | ---- |
> | Number | 2065 | 939      | 290  | 713   | 1495      | 1677      | 2041  | 9220 |
>
>
> Since SAR-AIRcraft-1.0 lacks diverse and separable imaging angles [R7-R9], a standard target recognition strategy and conducted experiments are implemented. Specifically,  80% of the data was used for training, and the remaining 20% for testing. The experimental results are outlined below:
>
>
> | Methods        | Overall Acc. (%) |
> | -------------- | ---------------- |
> | Resnet101      | 97.33±0.02       |
> | ΓPCA+Resnet101 | _**98.67±0.01**_ |
> | ViT-B/16       | 97.05±0.03       |
> | ΓPCA+ViT-B/16  | _**98.12±0.02**_ |
> | Swin-B         | 97.41±0.01       |
> | ΓPCA+Swin-B    | _**98.32±0.01**_ |
>
>
> Compared to the MSTAR dataset, SAR-AIRcraft-1.0 is lack of imaging angle annotations, and thus the angle-related constraints are eliminated in fact.  Consequently, in this experiment on SAR-AIRcraft-1.0, most of the evaluated networks exhibit superior performance. But notably, existing mainstream networks incorporating our ΓPCA consistently outperform their original backbones, even under this less constrained evaluation scenario. These results validate the generalizability of our method across diverse data characteristics and scenarios.
>
> **Reference:**
>
> [R3] Huang, Z., et al. Physics inspired hybrid attention for SAR target recognition. *ISPRS Journal of Photogrammetry and Remote Sensing*, 2024.
> [R4] Zhang, L., et al. Optimal azimuth angle selection for limited SAR vehicle target recognition. *International Journal of Applied Earth Observation and Geoinformation*, 2024.
> [R5] Wang, R., et al. MIGA-Net: Multi-view image information learning based on graph attention network for SAR target recognition. *IEEE Transactions on Circuits and Systems for Video Technology*, 2024.
> [R6] Li, W., et al. Hierarchical disentanglement-alignment network for robust SAR vehicle recognition. *IEEE Journal of Selected Topics in Applied Earth Observations and Remote Sensing*, 2023.
> [R7] Zhirui, W., et al. SAR-AIRcraft-1.0: High-resolution SAR Aircraft Detection and Recognition Dataset. *Journal of Radars*, 2023.
> [R8] Huang, B., et al. Scattering Enhancement and Feature Fusion Network for Aircraft Detection in SAR Images. *IEEE Transactions on Circuits and Systems for Video Technology*, 2024.
> [R9] Zhou, J., et al. DiffDet4SAR: Diffusion-based aircraft target detection network for SAR images. *IEEE Geoscience and Remote Sensing Letters*, 2024.
>
> ---
>
>
> > Suggestions and writing issues:
>
> **Answer2:**
> Thank you for your careful review and valuable suggestions. We will thoroughly proofread and incorporate these revisions into the final version of the paper.

---

### Official Review · Reviewer_8Si4 · 2025-03-16

**Overall Recommendation:** 3

**Summary:**

This paper proposes Gamma-PCA, a Gamma-distribution Principal Component Analysis model, to enhance angle-robust SAR target recognition by leveraging SAR's non-Gaussian statistics. The method derives consistent projection kernels to capture angle-invariant features without parameter updates, seamlessly integrating with CNN/Transformer backbones. Experiments on MSTAR demonstrate its effectiveness in mitigating performance degradation under significant azimuth/depression angle variations while maintaining computational efficiency.

**Claims And Evidence:**

Yes.

**Essential References Not Discussed:**

NA

**Experimental Designs Or Analyses:**

Yes.

**Methods And Evaluation Criteria:**

Yes.

**Other Comments Or Suggestions:**

NA

**Other Strengths And Weaknesses:**

see the comments.

**Questions For Authors:**

NA

**Relation To Broader Scientific Literature:**

NA

**Theoretical Claims:**

Yes.

---

> ### Author Rebuttal · Authors · 2025-04-01
>
> We sincerely appreciate your positive feedback on our work and are truly grateful for the time and effort you have dedicated to reviewing our manuscript. Should you have any additional comments or suggestions, we would be delighted to engage in further discussion and improvement.

---

> > ### Comment · Reviewer_8Si4 · 2025-04-07
> >
> > I have no more comments about the manuscript.

---

### Official Review · Reviewer_kgn7 · 2025-03-25

**Overall Recommendation:** 3

**Summary:**

This paper addresses a key challenge in SAR target recognition: the sensitivity of deep learning models to variations in azimuth and depression angles, which cause significant shifts in scattering characteristics. The authors propose a Gamma-Distribution Principal Component Analysis (ΓPCA) model to extract angle-invariant features by leveraging the statistical properties of SAR data. ΓPCA derives consistency convolution kernels without requiring parameter updates, thus adding no computational burden. The method is evaluated on the MSTAR dataset using ResNet and ViT backbones, demonstrating improved robustness to angle-induced distributional discrepancies. The key contributions include: (1) a novel ΓPCA framework tailored to SAR’s non-Gaussian statistics, (2) angle-insensitive feature extraction via a fixed projection matrix, and (3) seamless integration into existing architectures. While the approach shows promise, its generalizability to more diverse SAR targets and complex clutter scenarios remains to be further validated. The work stands out for its principled statistical modeling and parameter-free design, offering a potential advance over ad-hoc multiview fusion or data augmentation techniques.

**Claims And Evidence:**

The paper makes several strong claims, most of which are supported by empirical evidence, but some aspects could benefit from further clarification or validation.

**Essential References Not Discussed:**

N/A

**Experimental Designs Or Analyses:**

The experimental analyses are sound.

**Methods And Evaluation Criteria:**

To verify the effectiveness of the proposed method, the authors conducted a variety of robustness experiments on the MSTAR benchmark dataset. Experimental results show that the ΓPCA model can significantly improve the performance of existing models when facing angle changes. In addition, the ΓPCA model does not require parameter updates, so it does not bring additional computational burden to the network.

The proposed ΓPCA method and evaluation criteria (i.e., the MSTAR benchmark dataset) are meaningful for addressing the angle variation problem in SAR target recognition. The MSTAR dataset contains SAR images at different angles, which provides a suitable testing environment for evaluating the performance of the model at different observation angles. Therefore, it can be considered that the proposed model and evaluation method are designed for this problem or application and are reasonable.

**Other Comments Or Suggestions:**

N/A

**Other Strengths And Weaknesses:**

Strengths:

Strong motivation: Tackles a well-known but underexplored problem (angle sensitivity) in SAR ATR.

Novelty: ΓPCA extends PCA to Gamma distributions, aligning with SAR physics.

Practicality: No added parameters or training overhead.

Weaknesses:

Dependence on Gamma distribution assumptions (may not hold for all targets/clutter).

Limited discussion on computational efficiency vs. performance trade-offs.

**Questions For Authors:**

Please refer to the entries above.

**Relation To Broader Scientific Literature:**

The contribution of this paper is to propose a new feature extraction method that is particularly suitable for SAR data and can improve the robustness of the model in the face of angle changes. These contributions provide new perspectives and solutions compared to existing methods in the literature, especially in dealing with the non-Gaussian properties and angle changes of SAR data.

**Theoretical Claims:**

N/A, there is no theoretical claims in the paper.

---

> ### Author Rebuttal · Authors · 2025-04-01
>
> > Comment1. Dependence on Gamma distribution assumptions (may not hold for all targets/clutter).
>
> **Answer1:**
> We appreciate this insightful observation regarding the distributional assumption. To provide clarification:
> 1. Theoretical and empirical validation: The applicability of Gamma distribution to SAR image statistics is theoretically grounded and empirically validated, as evidenced by [R1, R2]. These works demonstrate its effectiveness in capturing the multiplicative speckle noise and scattering characteristics inherent to SAR systems.
> 2. Scope of the Gamma assumption: Our method adopts the Gamma distribution to model the reflection intensity of the entire SAR image (encompassing targets and clutter collectively). Of course, statistical distributions such as generalized Gaussian, fisher, etc. can be used, but these distributions are complex, and the corresponding parameter estimation and derivation of matrix $U$  are challenging to realize in practice.
> 3. Robustness of approach: By operating at this holistic statistical level, our method maintains robustness across diverse imaging scenarios, even when local target/clutter distributions vary. This aligns with established practices in SAR image analysis, where system-level statistical modeling often supersedes component-specific assumptions.
>
> **Reference:**
>
> [R1] Li, H. C., et al. On the empirical-statistical modeling of SAR images with generalized gamma distribution. *IEEE Journal of selected topics in signal processing*, 2011.
> [R2] Nascimento, A. D. C., et al. Compound truncated Poisson gamma distribution for understanding multimodal SAR intensities. *Journal of Applied Statistics*, 2023.
>
> ----
>
> > Comment2. Limited discussion on computational efficiency vs. performance trade-offs.
>
> **Answer2:**
> Thanks for your detailed comment. Below, we analyze the computational complexity of our proposed ΓPCA method and compare it with standard PCA.
>
> The primary computational cost of ΓPCA mainly arises from the derivation using the MM.
>
> 1) Update the mean vector $\mu^{(t+1)}$ by equation (13) in our manuscript:
>
> The update involves matrix multiplications and a linear system solve. Dominant operations include:
>
> - $O(d^2k)$ for $U^{(t)}(U^{(t)})^T$
>
> - $O(nd^2)$ for matrix multiplication of $H(·)(·)^T$
>
> Here, $n$ is the number of samples, $d$ is the original feature dimension and $k$ is the reduced dimension. Therefore, the total complexity for $\mu^{(t+1)}$ is $O(nd^2+d^2k)$.
>
>
> 2) Calculate the deviance $M(\boldsymbol{\Theta}|\boldsymbol{\Theta}^{(t)})$ by equation (27):
>
> Computes the Frobenius norm of an $n \times d$ matrix after rank-$k$ projections. Dominate terms:
>
> - $O(ndk)$ for matrix products with $\mathrm{H}+1_n\mu^{(t)}$ with $U^{(t)}$.
>
> - $O(d^{2}k)$ for $U^{(t)}(U^{(t)})^T$ operations.
>
>  The total complexity for $M(\boldsymbol{\Theta}|\boldsymbol{\Theta}^{(t)})$ is $O(ndk+d^2k)$.
>
> 3) Calculate the matrix $F^{(t)}$ by equation (30) and update $U^{(t+1)}$ to the first $k$ eigenvectors of $F^{(t)}$:
>
> - $O(nd^{2})$ for the matrix conjunction of $(·)^{T}Q^{(t)}(·)$.
> - $O(d^{3})$ for singular value decomposition (SVD).
>
> Therefore, the total complexity for $U^{(t+1)}$ is $O(nd^{2}+d^3)$.
>
> From the above discussion, the overall computational complexity of ΓPCA algorithm is approximately $O(nd^2+d^2k)+O(ndk+d^2k)+O(nd^2+d^3)\approx O(nd^2+d^2k+ndk+d^3).$
>
> In practical applications, parameters typically exhibit the following relationships: $n \gg d \gg k$. Therefore, the overall computational complexity of ΓPCA algorithm is $O(nd^2+d^3)$.
>
>
> In contrast, the computational complexity of the standard PCA method is mainly determined by two key operations:
>
> 1. Computation of covariance matrix: $\Sigma=\frac{1}{n}\mathbf{X}^T\mathbf{X}$, which has a computational complexity of $O(nd^2)$.
>
> 2. Eigenvalue decomposition of covariance matrix: $\Sigma=\mathrm{U}\Lambda\mathrm{U}^T$, with a computational complexity of $O(d^3)$, equivalent to the SVD operation in ΓPCA .
>
> Here, $\text{X}\in\mathbb{R}^{n\times d}$ is the data matrix, $U$ is the eigenvector matrix, and $\Lambda$ is the eigenvalue diagonal matrix. Consequently, the overall computational complexity of standard PCA is approximately $O(nd^2+d^3)$.
>
> The computational complexity is identical to that of standard PCA under the condition of $n \gg d \gg k$. The overall computational cost of our method is on the same order of magnitude as standard PCA, making the new algorithm computationally acceptable.

---

### Decision · Program_Chairs · 2025-05-01

**Decision:**

Accept (poster)

**Comment:**

This paper proposes a method to  address the sensitivity of deep learning models to variations in azimuth and depression angles in SAR data, which is termed  Gamma-Distribution Principal Component Analysis. After the rebuttal and discussion, three reviewers agree on the benefit of the approach in extending PCA to Gamma distributions, aligning with SAR physics, while one of the reviewers finds the theoretical and practical merits of ΓPCA not sufficiently validated and the novelty limited. While the AC agrees that the novelty is limited to this one aspect, the paper is still sufficiently motivated, theoretically sound, and it addresses an interesting problem from a signal processing perspective.